# Reversible two-way tuning of thermal conductivity in an end-linked star-shaped thermoset

Chase M. Hartquist [1,4], Buxuan Li [1,4], James H. Zhang[1,4], Zhaohan Yu [2], Guangxin Lv [1], Jungwoo Shin[1], Svetlana V. Boriskina [1], Gang Chen [1] ✉, Xuanhe Zhao [1,3] ✉ & Shaoting Lin [1,2] ✉

Polymeric thermal switches that can reversibly tune and significantly enhance their thermal conductivities are desirable for diverse applications in electronics, aerospace, automotives, and medicine; however, they are rarely achieved. Here, we report a polymer-based thermal switch consisting of an end-linked star-shaped thermoset with two independent thermal conductivity tuning mechanisms—strain and temperature modulation—that rapidly, reversibly, and cyclically modulate thermal conductivity. The end-linked star-shaped thermoset exhibits a strain-modulated thermal conductivity enhancement up to 11.5 at a fixed temperature of 60 °C (increasing from 0.15 to 2.1 W m$^{-1}$ K$^{-1}$). Additionally, it demonstrates a temperature-modulated thermal conductivity tuning ratio up to 2.3 at a fixed stretch of 2.5 (increasing from 0.17 to 0.39 W m$^{-1}$ K$^{-1}$). When combined, these two effects collectively enable the end-linked star-shaped thermoset to achieve a thermal conductivity tuning ratio up to 14.2. Moreover, the end-linked star-shaped thermoset demonstrates reversible tuning for over 1000 cycles. The reversible two-way tuning of thermal conductivity is attributed to the synergy of aligned amorphous chains, oriented crystalline domains, and increased crystallinity by elastically deforming the end-linked star-shaped thermoset.

Thermal switches are materials that change their thermal conductivities when subject to certain external stimuli, such as mechanical stresses, temperature changes, electrical fields, or magnetic fields[1,2]. High-performing thermal switches that can rapidly and reversibly tune their thermal conductivity are desirable in various modern technologies, including thermal management in electronic devices[3], temperature control in spacecrafts[4], and smart textile development for body temperature regulation[5,6]. Specifically, high-performing thermal switches remain pivotal in maintaining stable temperatures in spacecrafts, vehicles, infrastructures, and batteries amidst persistently fluctuating environmental conditions[1].

Existing efforts for developing thermal switches have leveraged electrochemical intercalation of layered materials[7–10], electric or magnetic field orientation of anisotropic heat-conducting particles[7], phase transition in phase change materials[11–13], and domain polarization in ferroelectric or ferromagnetic materials[3,14,15]. Despite the progress, these efforts still suffer drawbacks such as slow tuning speeds, poor cyclic performance, small tuning ratios, and specialized tuning modes. For example, electrochemical intercalation of layered materials

[1]Department of Mechanical Engineering, Massachusetts Institute of Technology, Cambridge, MA, USA. [2]Department of Mechanical Engineering, Michigan State University, East Lansing, MI, USA. [3]Department of Civil and Environmental Engineering, Massachusetts Institute of Technology, Cambridge, MA, USA. [4]These authors contributed equally: Chase M. Hartquist, Buxuan Li, James H. Zhang. ✉e-mail: gchen2@mit.edu; zhaox@mit.edu; linshaot@msu.edu

requires prolonged response times; electric or magnetic field orientation has a limited tuning ratio; and the thermal conductivity of phase-changing materials (e.g., ferroelectric and ferromagnetic materials) is typically discrete in switching mode. Therefore, there persists a strong need to develop new thermal switches to overcome these limitations and offer greater versatility and practicality for commercial applications.

Polymer-based thermal switches offer several advantages over conventional thermal switches: they are lightweight, flexible, easy to process, chemically stable, and mechanically durable[16–18]. Recent efforts have shown that polymer-based thermal switches can also have tunable thermal conductivity over a wide range by engineering polymer chain configurations[19–32], which enables better control over heat transfer and thermal management in various applications. However, there are still several limitations to these methods. For example, polyethylene films with an ultra-high thermal conductivity enhancement from 0.38 W m$^{-1}$ K$^{-1}$ at the as-extruded state (1×) to ~ 60 W m$^{-1}$ K$^{-1}$ at the highly drawn state (110×) have been reported[17,25,26]. While the thermal conductivity tuning ratio is as high as 160, this thermal conductivity change is irreversible and cannot be continuously tuned. While reversible tuning of thermal conductivity has been achieved through methods such as light-responsive azobenzene polymers[30] and tandem repetition proteins[31], the tuning modes in these methods are discrete, and the thermal conductivity cannot be continuously engineered. A recent breakthrough demonstrated reversible and continuous tuning of thermal conductivity in a crystalline polyethylene nanofiber by engineering the polymer chain through a temperature-induced structural phase transition[32]. However, this method is limited to micro-scale nanofibers due to challenges in engineering chain configurations in bulk-scale polymer samples. Some studies have shown that reducing entanglement in ultra-high-molecular-weight polyethylene tapes and films can result in high thermal conductivity[26], but this change is not reversible. Overall, although polymer-based thermal switches have great potential, further research remains vital to overcome these limitations and develop more practical and versatile thermal switch technologies.

In this work, we report a polymer-based thermal switch, employing an end-linked star-shaped thermoset (ELST) composed of tetra-arm polyethylene glycol (tetra-PEG). This ELST demonstrates a rapid, reversible, and cyclic modulation of thermal conductivity through a two-way tuning mechanism involving strain and temperature modulation. Specifically, the ELST exhibits a strain-modulated thermal conductivity tuning ratio up to 11.5 at a fixed temperature of 60 °C, increasing from 0.15 to 2.1 W m$^{-1}$ K$^{-1}$ when subjected to a change of stretch between 1 and 20. In addition, the ELST demonstrates a temperature-modulated thermal conductivity tuning ratio up to 2.3 at a fixed stretch of 2.5, increasing from 0.17 to 0.39 W m$^{-1}$ K$^{-1}$ when subjected to a change of temperature between 30 and 60 °C. The combined strain and temperature effects collectively enable the ELST to achieve a thermal conductivity tuning ratio up to 14.2. Notably, the thermal conductivity tuning of the ELST exhibits a shape-memory effect, through which the ELST can preserve its strain-modulated thermal conductivity through cooling to an unstressed state. Moreover, we also demonstrate the ELST's ability to maintain its thermal conductivity tuning reversibly and cyclically for 1000 cycles of loading. Through thermal–mechanical measurements and in situ X-ray characterizations, we show that the thermal transport mechanisms behind the two-way tuning of thermal conductivity are attributed to the synergy of aligned amorphous chains, oriented crystalline domains, and increased crystallinity in the end-linked star-shaped polymer network that possesses ultra-high stretchability and negligible trapped chain entanglements manifested in our recent work[33]. Molecular dynamics simulations quantify and validate thermal conductivity tuning at different levels of stretch, further highlighting the vital role of aligned amorphous chains in enhancing thermal

conductivity. This work unveils new strategies for engineering thermal conductivity in polymer systems with potential for applications in flexible thermal manipulation, such as solid-state refrigeration, thermal memory devices, and thermal metamaterials.

## Results

### End-linked star-shaped thermoset

The ELST is based on the A-B type tetra-arm polyethylene glycol (tetra-PEG) system, which was initially developed by Sakai et al.[34]. Unlike typical amorphous polymers with varying chain lengths and random topological defects, the A-B type tetra-PEG preserves a uniform chain length distribution with minimal topological defects by cross-linking two types of macromers in an equal molar ratio with a well-tuned reaction efficiency[35,36]. The nearly ideal-network polymers synthesized in this work are produced by combining the A-B type tetra-PEG macromers through static covalent bonds (e.g., NHS-amine), as demonstrated in prior works[34,36,37]. The molecular weights of type A and type B PEG macromers are set as 20,000 g/mol with each arm measuring 5000 g/mol. As previously reported[36], the reaction efficiency can be adjusted by deactivating the NHS-amine reaction: incubating the PEG macromers in an aqueous solution for a controlled time deactivates the NHS groups forming static covalent bonds. By manipulating the reaction efficiency, the occurrence of dangling-chain topological defects within the network can be controlled while reducing molecular entanglements and cyclic loops by adjusting the polymer concentration[36,38]. For this study, we set the reaction efficiency as a constant 0.93 to produce a nearly ideal tetra-PEG polymer network free of topological defects, leaving the study of the impact of topological defects as future work.

In contrast to previous studies primarily focused on tetra-PEG hydrogels[34,36,38], our study concentrates on solvent-free tetra-PEG thermosets[33], which exhibit a shape-memory effect with a distinct melting point (Supplementary Fig. S1), negligible trapped entanglements, and ultra-high stretchability when heated above $T_m$[33]. These unique mechanical and physical properties are attributed to the process of deswelling the end-linked star-shaped PEG macromers as manifested in our recent work[33]. As demonstrated in Fig. 1a and Supplementary Fig. S2, the tetra-PEG thermoset can undergo elastic deformation when heated above $T_m$ (e.g., $T = 60$ °C), maintain its temporary deformation at a fixed stretch ratio (e.g., $\lambda = \lambda^*$) upon cooling below $T_m$ (e.g., $T = 30$ °C), and elastically recover to its original shape when reheated above $T_m$ (e.g., $T = 60$ °C). To investigate the effect of strain on the material's thermal conductivity, we first prepare a series of tetra-PEG thermosets with controlled stretch ratios above $T_m$ (e.g., $T = 60$ °C) and then cool the samples to preserve their stretched states. To further investigate the effect of temperature on the material's thermal conductivity, we place a series of tetra-PEG thermosets with fixed stretch ratios within a chamber featuring controlled temperature settings and then conduct steady-state thermal conductivity measurements (Supplementary Information). We anticipate that the strain and temperature modulation induces a controlled transition between crystalline and amorphous states. As illustrated schematically in Fig. 1b, the tetra-PEG thermoset possesses a randomly dispersed semi-crystalline structure in its original undeformed shape ($\lambda = 1$) at a temperature below $T_m$ (e.g., $T = 30$ °C). When heated above $T_m$ ($T = 60$ °C), the tetra-PEG thermoset transforms into a stretchable, amorphous elastomer by melting the crystalline domains; therefore the elastomer can be elastically stretched to align the amorphous chains along the stretching direction. Due to the nearly ideal-network architecture in the tetra-PEG elastomer, we anticipate superior alignment of amorphous chains compared to conventional elastomers, which promotes efficient phonon transport, increasing thermal conductivity. Upon further cooling below $T_m$ (e.g., $T = 30$ °C), the tetra-PEG elastomer recovers its thermosetting behavior through cooling-induced crystallization. We ascertain the well-aligned chain

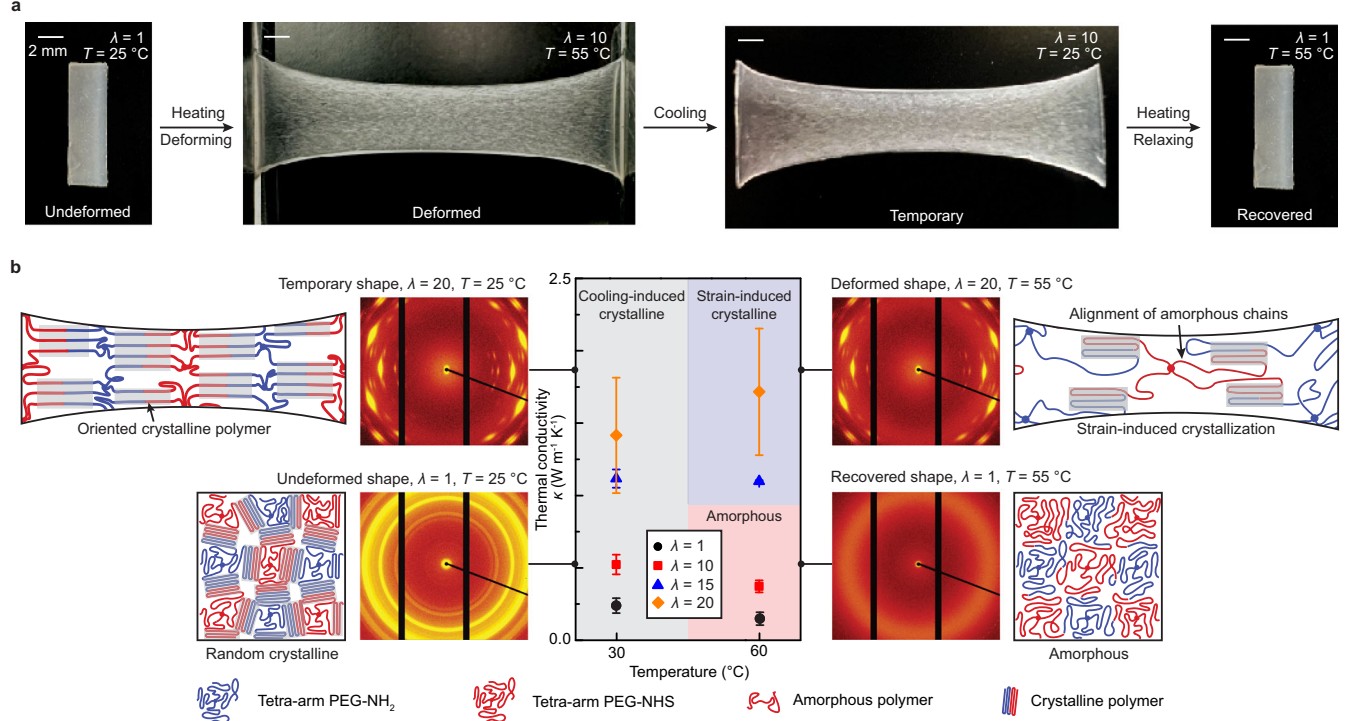

**Fig. 1 | Two-way Tuning of Thermal Conductivity in the ELST. a** Images of the shape-memory ELST from undeformed shape, deformed shape, temporary shape, to recovered shape with controlled strain and temperature modulation. **b** Schematics of thermal transport mechanism for enhanced thermal conductivity in the ELST by cooling-induced and/or strain-induced crystallization. Strain tunes thermal conductivity in the ELST by orienting crystalline domains, aligning polymer chains, and eliciting strain-induced crystallization. Temperature tunes by modulating the fractions of amorphous and crystalline regimes as avenues for phonon transport. Polymer orientation and chain alignment dominate in enhancing thermal conductivity. Schematics adapted from ref. **33**. © The Authors, some rights reserved, exclusive licensee AAAS. Distributed under a Creative Commons Attribution NonCommercial License 4.0 (CC BY-NC) http://creativecommons.org/licenses/by-nc/4.0/. Values in (**b**) represent the mean and the standard deviation (*n* = 3–5).

configuration in the stretched tetra-PEG elastomer leads to oriented crystalline domains and aligned interstitial amorphous chains, which significantly contributes to the enhancement of thermal conductivity. Conversely, heating the stretched tetra-PEG thermoset above $T_m$ ($T = 60$ °C) causes the oriented crystalline domains and aligned interstitial amorphous chains to return to their original randomly distributed semi-crystalline structure, leading to the material's original low thermal conductivity. Furthermore, the presence of crosslinkers in the tetra-arm PEG thermoset enables the reversible tuning of thermal conductivity through strain and temperature modulation.

## Thermal conductivity measurements

To study the thermal properties of ELST, we employ two distinct experimental schemes: a home-built steady-state system[17,39] and a frequency-domain thermoreflectance method (FDTR). Using the steady-state system, we measure the time-invariant heat flux by imposing steady-state temperature differences across a thin-film sample suspended between temperature-controlled hot and cold junctions (Supplementary Fig. S3). The system is placed in a vacuum chamber integrated with a turbomolecular pump (less than $5 \times 10^{-6}$ mbar) covered by a copper radiation shield. We analyze the systematic errors, including radiation and parasitic heat losses, and carefully minimize measurement errors through the stage design, differential protocol, and controlled experimentation[17] (refer to Supplementary Information for more details). The accuracy of the steady-state platform has been extensively validated by testing several control samples, including ultrathin polyethylene film, 304-stainless steel foil, Zylon fibers, Dyneema fibers, Sn films, and Al films with thermal conductivities ranging from 0.38 W m⁻¹ K⁻¹ to 202.7 W m⁻¹ K⁻¹, which have been shown in the previous literature[17]. We next measure the thermal

conductivity of a series of tetra-arm PEG thermosets with various stretch ratios and at different temperatures. We first investigate the impact of strain on thermal conductivity tuning. As shown in Fig. 2a, the undeformed ELST is found to have a thermal conductivity of 0.24 W m⁻¹ K⁻¹ at $T = 30$ °C (Supplementary Fig. S4). As the stretch ratio increases, the thermal conductivity of the deformed ELST along the stretch direction significantly increases, reaching 1.42 W m⁻¹ K⁻¹ at $T = 30$ °C. Notably, the ELST exhibits a maximum strain-modulated thermal conductivity tuning ratio up to 11.5 from 0.15 to 2.1 W m⁻¹ K⁻¹ at $T = 60$ °C (Fig. 2c). We further study the effect of temperature on thermal conductivity tuning. As shown in Fig. 2b, the rise in temperature slightly decreases the ELST's thermal conductivity at undeformed state (i.e., $\lambda = 1$). The maximum temperature-modulated thermal conductivity tuning ratio is 2.3 when the ELST is subjected to a fixed stretch of 2.5 (Fig. 2d). We interpret the slight drop in thermal conductivity with increasing temperature at $\lambda = 1$ is due to the melting of unoriented crystalline domains. Although significant change in the crystallinity occurs above 45 °C, we do not observe a large change in thermal conductivity because the thermal conductivity of polymers is governed predominantly by the amorphous regime[17]. Notably, as shown in Supplementary Fig. S4, at a fixed small stretch ratio (i.e., $\lambda = 2.5, 5$), the temperature shows negligible impacts on the on/off thermal conductivity tuning ratio; in contrast, at a fixed large stretch ratio (i.e., $\lambda = 10, 15, 20$), the temperature increase significantly enhances the on/off thermal conductivity tuning ratio $\kappa_{on}/\kappa_{off}$ up to 11.5. The nuanced variation in thermal conductivity in the deformed ELST at an elevated temperature is potentially a result of two competing phenomena: (1) the hindrance of phonon transport in the ELST due to the crystalline-to-amorphous transition, and (2) the augmentation of phonon transport in the ELST due to the aligned polymer

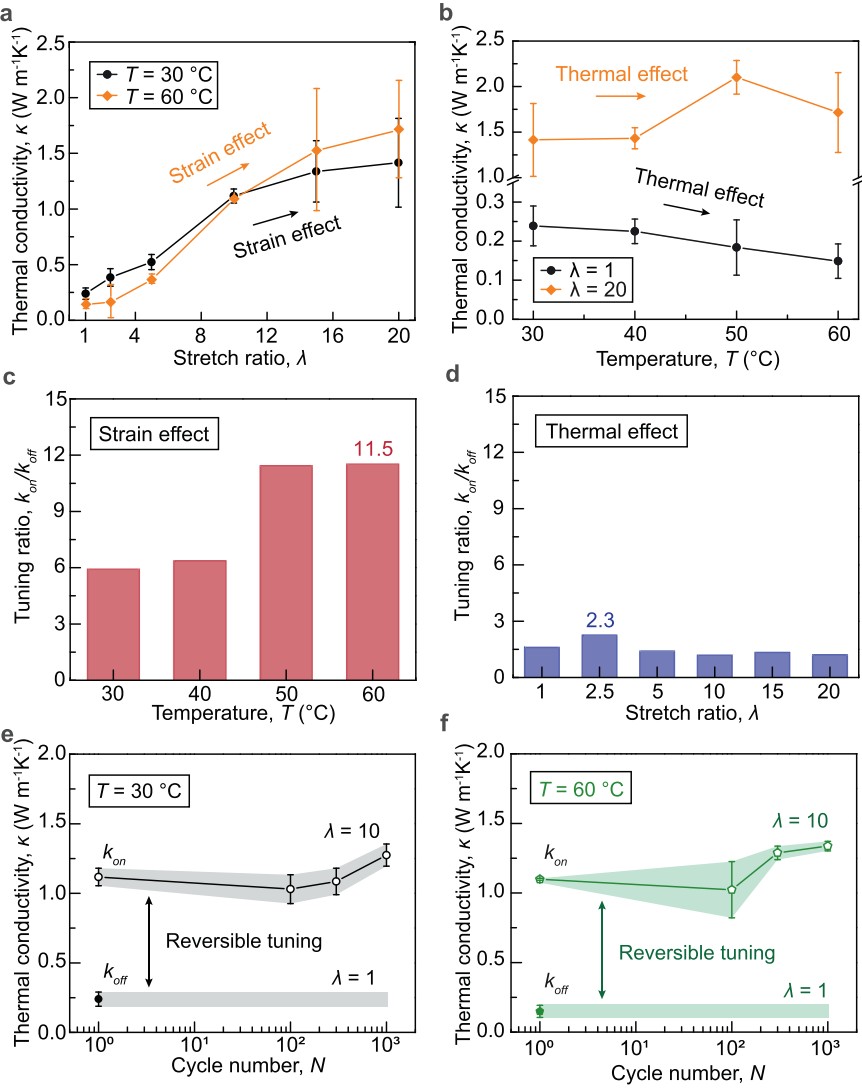

**Fig. 2 | Thermal conductivity characterization of the 20,000 MW ELST.**
**a** Thermal conductivity $\kappa$ versus stretch ratio $\lambda$ at fixed temperatures $T$ of 30 and 60 °C. **b** Thermal conductivity $\kappa$ versus temperature $T$ in the undeformed (i.e., $\lambda = 1$) and highly deformed (i.e., $\lambda = 20$) states. **c** Thermal conductivity on/off tuning ratio $\kappa_{on}/\kappa_{off}$ of the ELST at fixed temperatures $T$ of 30, 40, 50, and 60 °C. **d** Thermal conductivity on/off tuning ratio $\kappa_{on}/\kappa_{off}$ of the ELST at fixed stretch ratios of 1, 2.5, 5, 10, 15, 20. **e** Reversible tuning of thermal conductivity by cyclic stretch between $\lambda = 1$ and $\lambda = 10$ at $T = 30$ °C. **f** Reversible tuning of thermal conductivity by cyclic stretch between $\lambda = 1$ and $\lambda = 10$ at a temperature of $T = 60$ °C. Values in (**a**, **b**, **e**, **f**) represent the mean and the standard deviation ($n = 3–5$).

chains. It is important to highlight that the two-way thermal conductivity tuning of our ELST is primarily driven by strain modulation, with thermal modulation serving as an additional tuning capacity. Notably, employing single-field control (i.e., strain control) of our ELST can already achieve ~80% of the maximum tuning capacity, resulting in an 11.5× thermal conductivity tuning ratio. This underscores the feasibility of utilizing strain modulation as the primary control, complemented by thermal modulation for supplementary tuning, making the ELST well-suited for practical applications. While most thermal switches respond to a single stimulus, our ELST unveils avenues to explore how responses to multiple stimuli could introduce a new paradigm, which could potentially yield unprecedented results.

In addition to the steady-state system, we further employ the FDTR method to study the transient heat conduction in the tetra-PEG thermosets and further validate the steady-state results. As shown in Supplementary Fig. S5, we embed the bulk sample in an epoxy matrix and cut the sample perpendicular to the stretch direction using microtome equipment, creating an ultra-smooth surface for the FDTR measurement (refer to Supplementary Materials for more details). Representative thermoreflectance signals are reported in

Supplementary Fig. S5f, from which we extract the thermal conductivity of a series of tetra-PEG thermosets with different stretch ratios. The thermal conductivity of the undeformed tetra-PEG thermoset is measured to be 0.3 W m⁻¹ K⁻¹ at $T = 25$ °C. As the stretch ratio increases, the thermal conductivity of the deformed tetra-PEG thermosets along the stretch direction significantly increases, reaching 1.6 W m⁻¹ K⁻¹ at $\lambda = 20$ (Supplementary Fig. S6a). The FDTR results agree well with the measured values using the steady-state system.

We next measure the thermal conductivity of the ELST with controlled cycles of loading at various temperatures. Our data show that the thermal conductivity of the tetra-PEG thermosets with stretch ratios of 10 at the temperatures of $T = 30, 40, 50, 60$ °C remains constant even when the sample is subjected to 1000 cycles of loading (Fig. 2e, f and Supplementary Fig. S7), demonstrating its reversible and reliable tuning of thermal conductivity. Notably, we observe a change in thermal conductivity as the cycle number increases. This is due to the mechanical training that further increases crystallinity, orients crystalline domains, and aligns interstitial polymer chains, which matches results reported in a PVA hydrogel material with a similar semi-crystalline architecture[40]. While the thermal conductivity of the

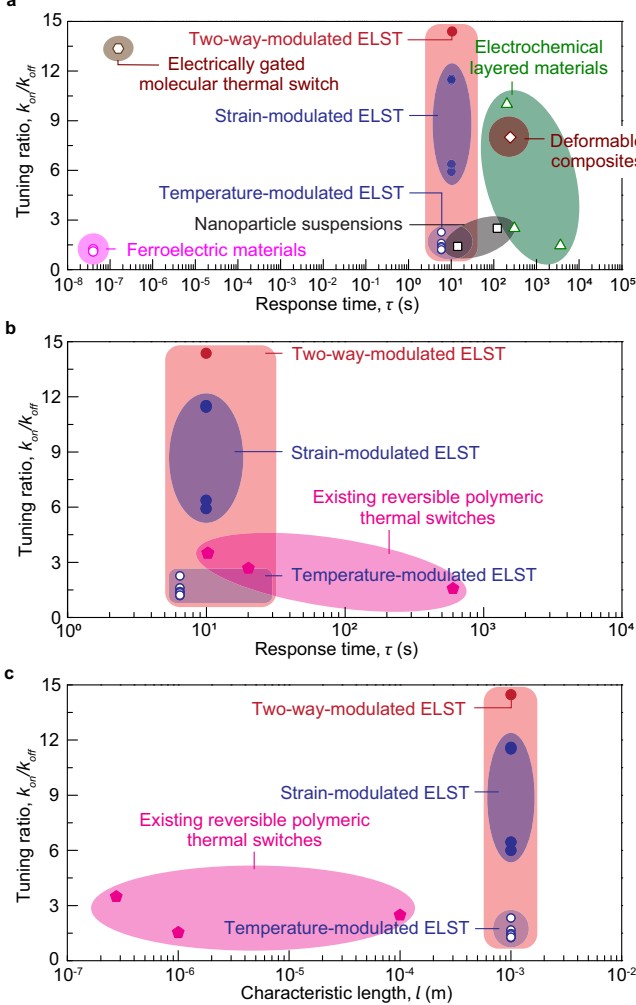

**Fig. 3 | Thermal conductivity tuning performances of the 20,000 MW ELST.**
**a** Comparison chart in the plot of thermal conductivity on/off tuning ratio $\kappa_{on}/\kappa_{off}$ versus response time $\tau$, comparing our ELST to existing reversible thermal switches such as an electrically gated molecular thermal switch[52], electrochemical layered materials[7,8,47], nanoparticle suspensions[48,65], ferroelectric materials[14,49], and deformable composites[3]. **b** Comparison chart in the plot of thermal conductivity on/off tuning ratio $\kappa_{on}/\kappa_{off}$ versus response time $\tau$, comparing our ELST to existing reversible polymeric thermal switches[30,31,51]. **c** Comparison chart in the plot of thermal conductivity on/off tuning ratio $\kappa_{on}/\kappa_{off}$ versus characteristic length $l$ (referring to the sample size that can be fabricated), comparing our ELST to existing reversible polymeric thermal switches[30,31,51].

ELST changes over cycle numbers at small cycles of loading, results suggest that the thermal conductivity should reach a steady-state value as manifested by its stress-stretch curves under cyclic loading (Supplementary Fig. S8). Preserving high thermal conductivity in polymers upon cyclic mechanical stretch is fundamentally challenging, because nearly all polymers including filled elastomers[41], interpenetrating hydrogels[42], and plastics[43] suffer fatigue[44], resulting in deteriorated molecular configurations with decreased thermal conductivity. Our recent work has shown that tetra-PEG hydrogels do not suffer fatigue manifested by negligible shakedown of stress over cycles (Supplementary Fig. S8)[36]. We ascertain that the tetra-PEG thermoset's ability to maintain high thermal conductivity over cycles of loading is possibly attributed to the uniform chain length and negligible topological defects in the nearly ideal polymer network, because the applied stress can be uniformly transferred to individual polymer chains, favoring the orientation of crystalline domains and alignment of interstitial amorphous chains. Notably, the reversible and

significant thermal conductivity tuning capabilities of our ELST have not been achieved in existing polymeric thermal switches.

We further estimate the response time for the two-way tuning of thermal conductivity in the ELST, a critical characteristic of thermal switches. The total response time of the two-way tuning of thermal conductivity includes two parts: the time for mechanical or thermal modulation and the response time for structural relaxation. Since the ELST's thermal conductivity is primarily affected by strain (Fig. 2a, c), we particularly focus on the time for mechanical modulation, which is necessary for large thermal conductivity tuning ratios up to 11.5. The time for mechanical modulation is determined by the time for stretching the material, which can be estimated by $\tau_{strain} = (\lambda - 1)L/V$, where $\lambda$ is the stretch ratio, $L$ is the sample's gage length, and $V$ is the loading speed. Given $\lambda = 20$, $L = 2$ mm, and $V = 5$ mm/s, the time for stretching the material is estimated as $\tau_{strain} = 7.2$ s. The time for thermal modulation is mainly dominated by the time for thermal conduction, which can be estimated by $\tau_{heat} = t^2/D_{heat}$, where $t$ is the sample thickness and $D_{heat}$ is the thermal diffusivity of the sample. Given $t = 1$ mm and $D_{heat} = 0.36$ mm²/s[45], the time for thermal conduction is estimated as $\tau_{heat} = 2.8$ s. We further perform photoelasticimetry experiments (Supplementary Fig. S9)[46] to quantify the response time for structural alternation in the ELST subjected to a nearly instantaneous stretch (i.e., a stretch of 4.4 in 1 s, Supplementary Fig. S10a). Specifically, from the measured nominal stress as a function of time, we extract the mean response time for the structural change of the entire sample as 2.58 s (Supplementary Fig. S10c); from the measured intensities at specific locations as a function of time, we find the response time for the structural change at these locations are consistently on the order of 1 s (Supplementary Fig. S10b, d). As summarized in Supplementary Fig. S10e, the response time for structural alternation of the ELST sample is orders of magnitude shorter than that of conventional polymers (e.g., 100 s in polyacrylamide gel). The short response time for structural relaxation in the ELST is attributed to its unique low topological defects and negligible molecular entanglements as demonstrated in our recent paper[33]. Considering both the time scale of thermal and mechanical stimuli, and the structural response time, we estimate that the switching time to be ~ 10 s. We also perform the cyclic relaxation experiments (Supplementary Fig. S11) to measure both the on and off tuning time as 2.58 s and 0.08 s, respectively. Notably, the time for off-tuning is shorter compared to on-tuning, which is possibly due to minimal alterations in the polymer-network architecture when our ELST is unloaded. Additionally, our data suggest that the on and off tuning time remain nearly identical across various cycles. Figure 3a summarizes the comparison chart in the plot of thermal conductivity on/off tuning ratio $\kappa_{on}/\kappa_{off}$ versus response time $\tau$. Our ELST exhibits a thermal conductivity on/off tuning ratio up to 11.5 at a fixed temperature of $T = 60$ °C, up to 2.3 at a fixed stretch of 2.5, up to 14.2 through a combined strain and temperature modulation, and response time on the order of 10 s, outperforming most existing thermal switches such as electrochemical layered materials[7,8,47], nanoparticle suspensions[48], ferroelectric materials[14,49], and deformable composites[3,50]. Particularly, when compared to existing reversible polymeric thermal switches[30,31,51], our ELST demonstrates a substantially improved tuning ratio and notably shortened tuning time (Fig. 3b), while preserving these features on a bulk scale (Fig. 3c and Supplementary Table 1). It should be noted that the recently reported electrically gated solid-state thermal switch[52] exhibits an ultra-short response time up to $10^{-6}$ s while maintaining an ultra-high thermal conductivity tuning ratio up to 13, significantly pushing the limit of existing thermal switches, including our ELST. However, such fast modulation so far is on a monolayer at the interface region. In contrast, our ELST can be fabricated on a bulk scale through mass production.

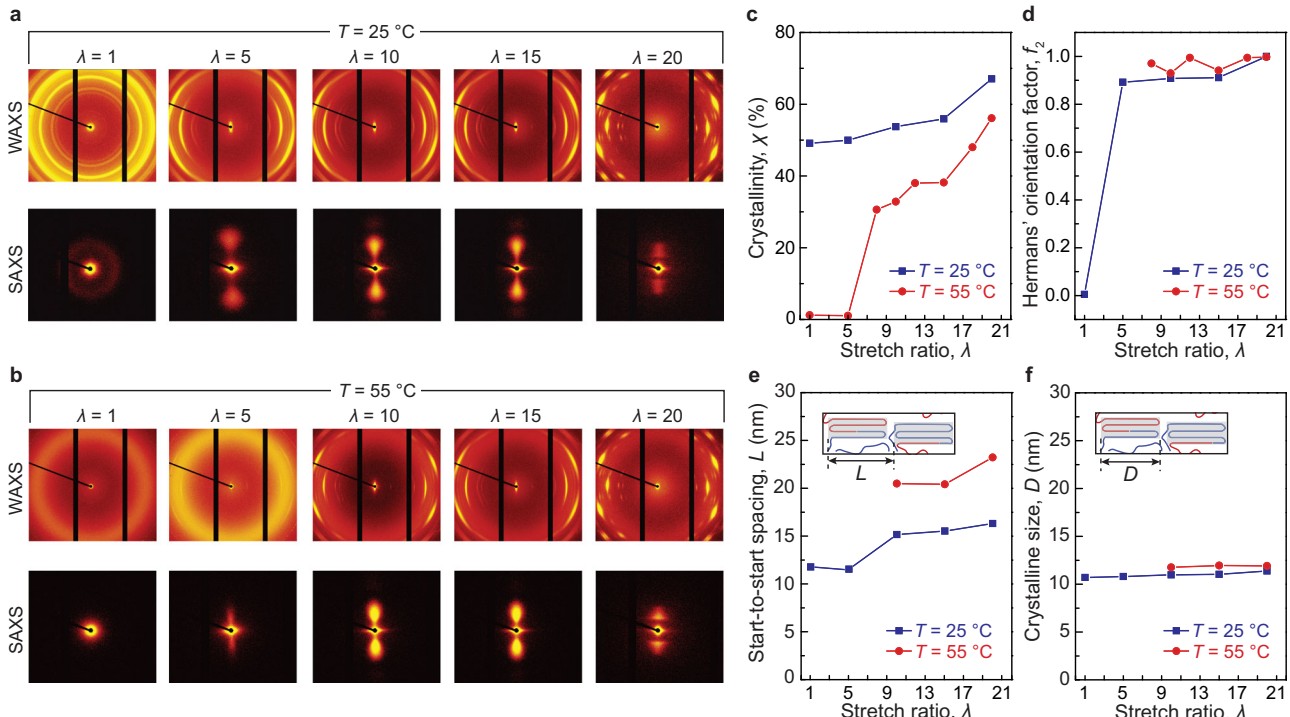

**Fig. 4 | The strain effect on ELST structure characterized via X-ray scattering.**
**a** Small- and wide-angle scattering pattern of the ELST at different stretch ratios at
$T = 25\,°C$. **b** Small- and wide-angle scattering patterns of the ELST at different
stretch ratios at $T = 55\,°C$. **c** Crystallinity $\chi$, **d** Hermans' orientation factor $f_2$, **e** Start-
to-start spacing $L$, and **f** crystalline domain size $D$ versus stretch ratio $\lambda$ at $T = 25$
and $55\,°C$.

## Thermal transport mechanism

Polymers are typically classified as thermal insulators and display
thermal conductivities on the order of $0.2\,W\,m^{-1}\,K^{-1}$ at room tem-
perature because localized vibrational modes dominate due to strong
heterogeneous inter-/intra-chain interactions, and the mean free path
of heat-conducting phonons reduces due to the random arrangement
of amorphous chains. Recent efforts show that the thermal con-
ductivity of polymers can be significantly enhanced up to
~$4.4\,W\,m^{-1}\,K^{-1}$ for polythiophene nanofibers[29], $51–104\,W\,m^{-1}\,K^{-1}$ for
polyethylene nanofibers[24,27,53], and $62–65\,W\,m^{-1}\,K^{-1}$ for polyethylene
films[17,26] through alignment of amorphous chains, orientation of crys-
talline domains, and increase of crystallinity. The most common
method to increase thermal conductivity in polymers is via thermal
drawing of uncrosslinked polymers, which unfortunately prevents
reversible tuning of thermal conductivity. Introducing crosslinks into
polymers can reversibly tune thermal conductivity by elastically
deforming the crosslinked polymer networks, but the tuning ratio of
thermal conductivity is very low due to the presence of crosslinks,
topological defects, and chain heterogeneity, which act as stress
concentration points and phonon scattering sites for heat transfer[54].

We hypothesize that the synergy of oriented crystalline domains,
aligned interstitial amorphous chains, and increased crystallinity in the
nearly ideal polymer network explains the two-way tuning of thermal
conductivity observed in the ELST (Supplementary Fig. S12). Specifi-
cally, at a moderate stretch ratio, the enhanced phonon transport is
mainly due to the oriented crystalline domains; as the stretch ratio
increases, the crystalline domains are further oriented and the inter-
stitial amorphous chains are well-aligned along the stretch direction,
further enhancing thermal transport. To validate our hypothetic
thermal transport mechanism, we perform systematic X-ray scattering
characterizations to quantify the size, density, orientation, and align-
ment of crystalline domains and interstitial amorphous chains in a
series of ELSTs with various stretch ratios and at different tempera-
tures (Supplementary Fig. S13). Small-angle X-ray scattering (SAXS)

and wide-angle X-ray scattering (WAXS) are performed using a Dectris
Pilatus3R 300 K detector on a SAXSLAB apparatus[33].

We use X-ray characterization to investigate strain as the first
mechanism for thermal conductivity tuning (Fig. 4 and Supplementary
Fig. S14). The ELSTs are stretched and cooled to room temperature and
fixed on both ends with Krazy glue to an acrylic mount (See details in
Supplementary Information). Figure 4a, b plots the small-angle X-ray
scattering (SAXS) and wide-angle X-ray scattering (WAXS) patterns of
the samples under different stretch ratios at a fixed room temperature
of 25 and 50 °C, respectively. We first employ the measured WAXS
intensity profile to quantify the crystallinity in the undeformed ELST
(Fig. 4c). As shown in Supplementary Fig. S14c, d, the presence of
narrow peaks denotes the formation of crystalline domains. The
crystallinity of the sample can be quantified by fitting the measured
WAXS intensity distributions with Gaussian or Pseudo-Voight func-
tions (Supplementary Fig. S16). As shown in Fig. 4c, the measured
crystallinity of the undeformed nearly ideal-network polymer at
$T = 25\,°C$ is measured to be 48%. As the stretch ratio increases, the
crystallinity of the deformed ELST gradually increases up to 66%,
indicating a pronounced increase of crystallinity as stretch ratio
increases. Next, we analyze the azimuthal spread of diffraction peak
intensity to characterize the orientation of crystalline domains. The
orientation order of crystalline domains is assessed using Hermans'
orientation factor, defined as $f_2 = (3\langle\cos^2\phi\rangle - 1)/2$, where $\phi$ is the
azimuthal angle (Supplementary Fig. S17). As shown in Fig. 4d, the
undeformed ELST shows nearly zero Hermans' orientation factor,
suggesting the nature of randomly distributed crystalline domains. As
the stretch ratio reaches 5, the Hermans' orientation factor sig-
nificantly increases to 0.9, validating that oriented domains dominate
in the tetra-PEG thermoset at moderate deformations. As the stretch
ratio increases, the Hermans' orientation factor further increases up to
1.0, suggesting that crystalline domains become almost fully oriented
along the stretch direction. Given the measured WAXS and SAXS data,
we further quantify the size of crystalline domains $D$ and start-to-start

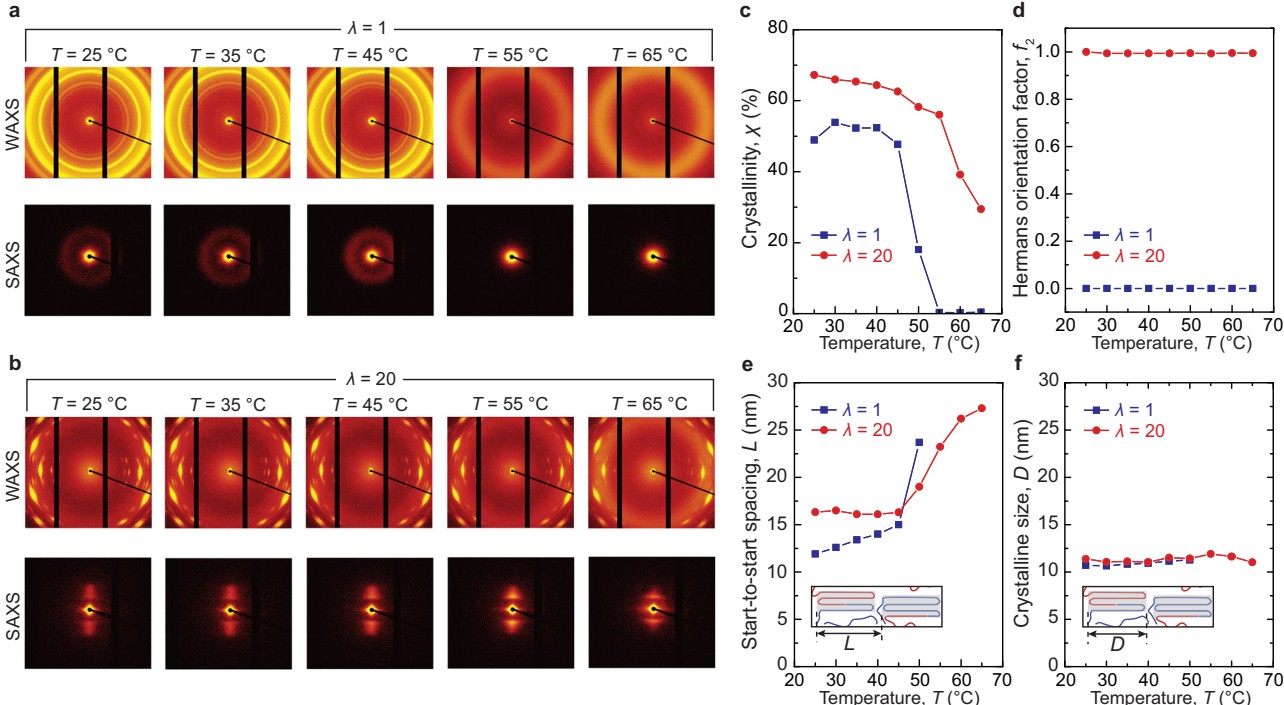

**Fig. 5 | The temperature effect on ELST structure characterized via X-ray scattering. a** Small- and wide-angle scattering patterns of the ELST at different temperatures at undeformed state (i.e., $\lambda = 1$). **b** Small- and wide-angle scattering patterns of the ELST at different temperatures at highly deformed state (i.e., $\lambda = 20$). **c** Crystallinity $\chi$, **d** Hermans' orientation factor $f_2$, **e** Start-to-start spacing $L$, and **f** crystalline domain size $D$ versus temperature $T$ at $\lambda = 1$ and 20.

spacing $L$ between adjacent crystalline domains for the ELST at various stretch ratios. As summarized in Fig. 4f, the size of crystalline domains remains constant at around 12 nm, while the distance between repeating structural units of crystalline and amorphous domains gradually increases from 11.9 to 16.3 nm as the stretch ratio increases (Fig. 4e). The preservation of crystalline domains justifies the consistent crystalline domain size as the ELSTs are heated, stretched, and cooled; furthermore, the increased distance between crystalline domains indicates the alignment of amorphous chains between crystalline domains.

At $T = 55\,°C$ above $T_m$, we observe an intriguing strain-induced crystallinity up to 50% when stretched to 20 (Fig. 4b, c), significantly outperforming natural rubber (i.e., 14% at 22 °C and 10% at 55 °C) and other well-studied elastomers including butyl rubber, polyisoprene rubber, and polybutadiene rubber[33]. In addition, we find that the temperature rise can further increase the Hermans' orientation factor of the ELST, indicating a further orientation of crystalline domains at $T = 55\,°C$ (Fig. 4d). Furthermore, since the polymer chain is more flexible at $T = 55\,°C$, the start-to-start distance spacing $L$ between adjacent crystalline domains increases from 20 to 23 nm as the stretch ratio increases, larger than that at $T = 25\,°C$ (Fig. 4e). Notably, the size of crystalline domains $D$ of the ELST at $T = 55\,°C$ remains almost the same as that at $T = 25\,°C$ (Fig. 4f). The ultra-high strain-induced crystallinity, the further increased orientation of crystalline domains, and the enlarged start-to-start spacing between adjacent crystalline domains in the ELST possibly explains the enhanced thermal conductivity tuning ratio up to 11.5 at an elevated temperature.

We further use X-ray characterizations to study temperature modulation as the second thermal conductivity tuning mechanism (Fig. 5 and Supplementary Fig. S15). Figure 5a plots the SAXS and WAXS scattering patterns of the ELST under different temperatures at an undeformed state ($\lambda = 1$). As shown in Fig. 5c, the crystallinity of ELST decreases significantly with rising temperature, attributed to the melting of crystalline domains. Despite the significant change in the crystallinity that occurs above $T_m$, the ELST's thermal conductivity

exhibits a slight drop because the thermal conductivity of polymers is governed predominately by the amorphous regime[17]. The undeformed ELST maintains an almost zero Hermans' orientation factor irrespective of temperature, indicating that temperature does not promote the orientation of crystalline domains. Furthermore, the start-to-start spacing $L$ between adjacent crystalline domains increases as the temperature increases, which is due to the melting of crystalline domains that make crystalline domains sparser (Fig. 5e). In addition, Fig. 5f reveals no temperature dependency on crystalline size. Similarly, when ELST is highly deformed (i.e., $\lambda = 20$), the ELST shows a decrease in crystallinity (Fig. 5c), an augmentation in start-to-start spacing (Fig. 5e), a constant Hermans' orientation factor, and an unaltered crystalline size as the temperature increases.

## Molecular dynamics simulation

Beyond the X-ray scattering characterizations, we carry out all-atom molecular dynamics (MD) simulations to model the change in thermal conductivity at different stretch ratios in the 10,000 molecular weight (MW) tetra-PEG networks. Many previous MD simulation efforts have been made to understand the effect of mechanical deformation on thermal conductivity in polymers[55–57]. As illustrated in Supplementary Fig. S18, we first generate a series of simulated end-linked star-shaped thermosets with various controlled stretch ratios. A fully extended diamond lattice of the ELST is first initialized in the simulation cell at $T = 596\,K$ and collapsed isotropically at a constant rate of 5 nm/ns in each axis to about 25% of the final density at room temperature. The collapsed polymer network is then cooled to $T = 298\,K$ at a rate of 20 K/ns and equilibrated at 298 K and 1 atm for 5 ns in the constant-temperature, constant-volume (NPT) ensemble, creating the ELST at the undeformed state. To create the simulated ELST at various stretch ratios, we heat the polymer network to $T = 353\,K$ at a rate of 20 K/ns and then equilibrate it for 1 ns. The heated ELST is then stretched in the $x$-axis at a constant engineering strain rate of 0.5 ns$^{-1}$. At each specified stretch ratio, the simulated polymer network is snapshotted, cooled to $T = 298\,K$ at a rate of 20 K/ns, and equilibrated for 5 ns for subsequent

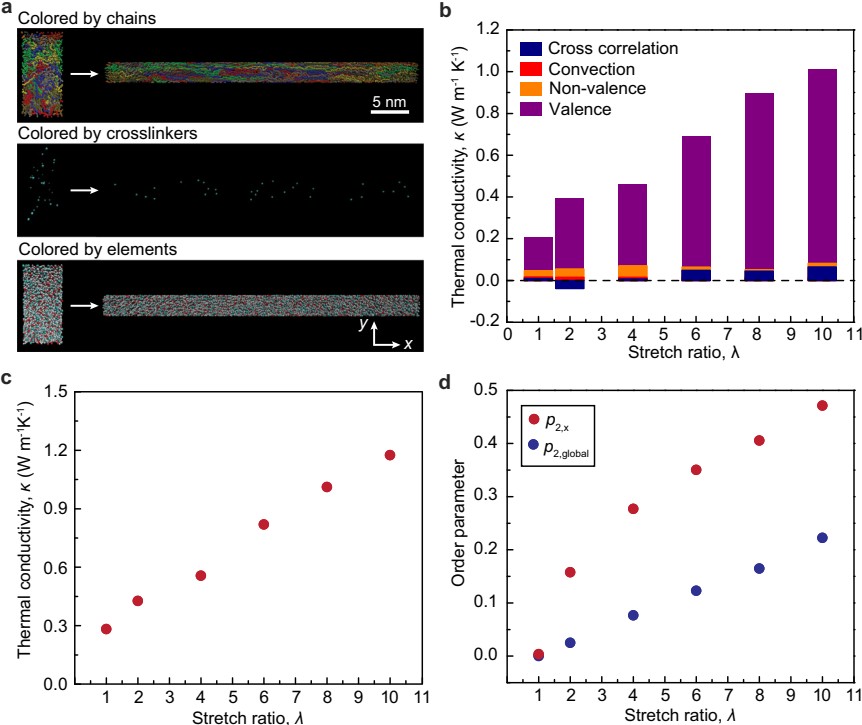

**Fig. 6 | Molecular dynamics simulation of thermal conductivity in the 10,000 MW ELST. a** Snapshots of trajectory showing simulated ELST at undeformed and deformed states colored by chains (top), crosslinkers (middle), and elements (bottom). **b** Four contributions to the total thermal conductivity: convection autocorrelation, non-valence autocorrelation, valence autocorrelation, and cross-correlation terms. **c** Simulated thermal conductivity as a function of stretch ratio. **d** Calculated order parameters, $p_{2,x}$ and $p_{2,\text{global}}$, to characterize the local chain alignment with stretch direction and global crystalline orientation along the stretch direction, respectively.

structural characterization and thermal conductivity calculation. Figure 6a shows a snapshot of the simulated ELST at undeformed and highly deformed states, colored by chains, crosslinks, and elements.

To calculate the thermal conductivity for each simulated sample, we use the Green–Kubo method for calculation. Specifically, given the calculated heat flux vectors **J** at every timestep, the thermal conductivity is equal to[58]

$$\kappa = \frac{V}{k_{\text{B}}T^2} \int_0^\infty \langle \mathbf{J}(t)\mathbf{J}(0) \rangle dt \qquad (1)$$

where $\mathbf{J}(t)$ is the heat flux vector at a given time $t$, $V$ is the system volume, and $T$ is the system temperature. The total thermal conductivity can be decomposed into four contributions: convection autocorrelation $\kappa_{\text{conv-conv}}$, non-valence autocorrelation $\kappa_{\text{nonval-nonval}}$, valence autocorrelation $\kappa_{\text{val-val}}$, and cross-correlation $\kappa_{\text{cross}}$ terms, namely,

$$\kappa = \kappa_{\text{conv-conv}} + \kappa_{\text{nonval-nonval}} + \kappa_{\text{val-val}} + \kappa_{\text{cross}} \qquad (2)$$

As shown in Fig. 6b, the valence autocorrelation to the total thermal conductivity dominates the enhanced thermal transport as the stretch ratio increases. Figure 6c plots the calculated thermal conductivity in simulation for the 10,000 MW tetra-PEG thermoset at different stretch ratios. As the stretch ratio increases, the total thermal conductivity increases accordingly, motivating the critical role of aligned interstitial amorphous chains in enhancing the ELST's thermal conductivity.

We further quantify the structural changes as the stretch ratio increases to understand the effects of crystalline orientation and chain

alignment on the ELST's thermal conductivity. We calculate two order parameters, $p_{2,x}$ and $p_{2,\text{global}}$, to characterize the overall chain alignment and global crystalline orientation along the stretch direction, respectively. The calculations of the two order parameters are expressed as:

$$p_{2,x} = \left\langle \frac{2}{3}\cos^2\theta_{i,x} - \frac{1}{2} \right\rangle_i \qquad (3-1)$$

$$p_{2,\text{global}} = \left\langle \frac{3}{2}\cos^2\theta_{i,j} - \frac{1}{2} \right\rangle_{ij} \qquad (3-2)$$

where $\theta$ is the angle between two vectors, $i$ represents the vector connecting the $(i-1)$ atom to the $(i+1)$ atom along the chain backbone, $j$ represents the vector connecting the $(j-1)$ atom to the $(j+1)$ atom along the chain backbone, and $x$ represents the cartesian axis along the stretch direction. As shown in Fig. 6d, the order parameter $p_{2,\text{global}}$ displays a roughly linear increase from 0 to 0.22, indicating an increase in the overall crystallinity due to polymer chain alignment. This is further supported by the chain alignment order parameter along the stretching direction, $p_{2,x}$, which initially shows a large increase and further raises to 0.47 at a stretch ratio of 10. The increasing trend of the order parameter with the increasing stretch qualitatively matches our X-ray measurements.

## Discussion
In summary, this work reports on a new class of polymeric thermal switches made of an end-linked star-shaped thermoset (ELST) composed of tetra-arm PEG polymers. This ELST demonstrates rapid, reversible, and cyclic modulation of thermal conductivity through a

two-way tuning mechanism involving strain and temperature modulation. Specifically, the ELST exhibits a strain-modulated thermal conductivity tuning ratio up to 11.5 and a temperature-modulated thermal conductivity tuning ratio up to 2.3. The combined strain and temperature effects collectively enable the ELST to achieve a thermal conductivity tuning ratio up to 14.2. We further demonstrate the ELST's ability to maintain its thermal conductivity tuning reversibly and cyclically for 1000 cycles of loading. Notably, the thermal conductivity tuning of the ELST exhibits a shape-memory effect, through which the ELST can preserve its strain-modulated thermal conductivity through cooling to an unstressed state. Our study, supported by thermal-mechanical measurements and in situ X-ray characterizations, reveals that the thermal transport mechanisms responsible for the two-way tuning of thermal conductivity rely on the synergy of aligned amorphous chains, oriented crystalline domains, and increased crystallinity within the ELST, which possesses ultra-high stretchability and contains negligible trapped chain entanglements. Quantitative validation through molecular dynamics simulations further highlights the vital role of aligned amorphous chains in enhancing thermal conductivity at different levels of stretch. These findings uncover new strategies for engineering thermal conductivity in polymer systems, exposing a breadth of potential applications in flexible thermal manipulation such as solid-state refrigeration, thermal memory devices, and thermal metamaterials. For example, our ELST's two-way tuning of thermal conductivity potentially expands engineering spaces for developing the next generation of elastocaloric cooling materials, which allows us to push the boundaries of elastocaloric cooling performance while minimizing dissipation to the surrounding environment.

## Methods
### Materials
In total, 20,000 MW tetra-arm amine-terminated PEG (Laysan Bio, 4 arm PEG-NH$_2$, MW 20,000) and 20,000 MW tetra-arm NHS-terminated PEG (Laysan Bio, 4 arm PEG-SG, MW 20,000) are the two types of macromers used for synthesizing 20,000 MW ELST. Phosphate buffered saline (Sigma-Aldrich, P4417) is used as the buffer to dissolve amine-terminated PEG, and a phosphate-citrate buffer (Sigma-Aldrich, P4417) is used to dissolve NHS-terminated PEG. In preparing the sample for FDTR thermal conductivity measurement, the EMbed 812 Embedding Kit (Electron Microscopy Sciences) is used to embed the ELSTs in preparation for microtome cutting.

### Synthesis of the ELST
We synthesize the tetra-PEG hydrogel following the protocol reported in past work[36,37], and then make it into the ELST via the dehydration process. 100 mg of tetra-arm amine-terminated PEG is first dissolved and vigorously mixed in a 1 mL phosphate buffer solution (one tablet dissolved in 300 mL deionized water), yielding a pH of 7.4 and ionic strength of 100 mM. Thereafter, 100 mg of tetra-arm amine-terminated PEG is dissolved and vigorously mixed in 1 mL of a phosphate-citrate buffer solution (one tablet dissolved in 150 mL deionized water), yielding a pH of 5.8 and ionic strength of 100 mM. Both the solutions of PEG-NH$_2$ and PEG-NHS are vigorously mixed and poured into an acrylic mold, giving a final concentration of 50 mg/mL for both PEG-NH$_2$ and PEG-NHS macromers. The resultant samples are placed in a humidity chamber for at least 12 h to complete the reaction of forming amide bonds between macromers. To obtain the solvent-free tetra-PEG thermoset, the as-prepared tetra-PEG hydrogel is placed in an environmental chamber at 37 °C for subsequent dehydration. To enable uniform and isotropic shrinkage of ideal-network PEG hydrogels, a thin layer of silicon oil is introduced at the interface between the hydrogel and the substrate to mitigate interfacial adhesion. The complete dehydration process typically takes about 12 h. Samples are carefully heated above $T_m$ and cooled to room temperature to alleviate

any residual stress buildup. More details on the preparation of the ELST with controlled stretch ratios and the ELST for FDTR thermal measurements are in the Supplementary Text.

### Home-built steady-state system
We use a home-built steady-state differential thermal conductivity stage to measure the thermal conductivity of ELSTs along the stretching direction (Supplementary Fig. S3a). Using the steady-state system, we measure the time-invariant heat flux given a set of constant temperature differences across the sample. The sample is mounted between a hot junction and a cold junction, which are connected to a temperature-controlled heater and a thermoelectric cooler (TEC), respectively. The TEC dissipates heat to a water-cooled heat sink. Thermocouples are connected to the heater and a cold junction, whose temperatures are measured. The heat flux measured as the electrical heating power of heater ($P_e$) is monitored by measuring the imposed voltage and current. The platform is surrounded by a copper radiation shield, whose temperature is measured by a thermocouple and controlled at the same temperature as the hot junction via a heater. The radiation shield is thermally insulated from the water-cooled heat sink by cylindrical porous ceramic spacers. The whole platform is put into a vacuum chamber (less than $5×10^{-6}$ mbar with a turbomolecular pump). With the determination of sample geometry (i.e., width $w$, length $L$, and thickness $t$), temperature difference between hot and cold junctions ($T_h - T_c$), and heat flux $Q$, we can use the Fourier law of heat conduction to extract the sample thermal conductivity via $Q = kwt/L(T_h - T_c)$. More details on the thermal conductivity measurement using the home-built steady-state system are in the Supplementary Text.

### Frequency-domain thermoreflectance method
We also use a frequency-domain thermoreflectance (FDTR) to measure the thermal conductivity of ELSTs, following the method developed by Schmidt et al.[59–62] The FDTR platform consists of two continuous-wave lasers: a pump laser with a wavelength of 488 nm and a probe laser with a wavelength of 532 nm. The pump laser modulated sinusoidally from 3 kHz to 10 MHz serves to heat the sample, while the probe laser serves to detect the surface temperature of the sample via the thermoreflectance effect of the coated transducer layer (Supplementary Fig. S5d). The phase of the pump beam at each modulation frequency is first determined before the FDTR measurement. Thereafter, we record the phase lag between the modulated surface temperature and the sinusoidal FDTR signal. The measured FDTR phase lag data is further fitted to an isotropic two-layer analytical model, using the sample thermal conductivity and the Au-sample interfacial conductance as fitting parameters[59–62]. More details on the thermal conductivity measurement using the frequency-domain thermoreflectance method are in the Supplementary Text.

### Small- and wide-angle X-ray scattering
Small-angle X-ray scattering (SAXS) and wide-angle X-ray scattering (WAXS) are performed using X-ray Diffraction Facility at Massachusetts Institute of Technology (i.e., Dectris Pilatus3R 300 K detector on a SAXSLAB apparatus). To reduce the background intensity fluctuation, the vacuum chamber is pumped to 0.08 mbar during the X-ray scattering measurements. Details on the measurement configurations are listed in Supplementary Table S2. Tensile specimens (~4 mm × 1 mm × 0.6 mm) are stretched for bulk structural characterization using the same procedure used for the thermal conductivity measurement. The stretched specimens are cooled to room temperature and fixed to acrylic fixtures using Krazy glue. We fit the measured 1D intensity profile into Gaussian or Pseudo-Voigt curves, extracting the area under the crystalline peaks $A_C$ and the area under amorphous peaks $A_A$ after subtracting the background scattering intensity, which determines the crystallinity as $\chi = A_C/(A_C + A_A)$. The Hermans' orientation parameter $f_2$

is determined via $f_2 = (3\langle \cos^2\phi \rangle - 1)/2$, where $\phi$ is the azimuthal angle. The crystalline size can be extracted via the Scherrer equation, $D = K\Lambda/(B(2\theta)\cos\theta)$, where $K$ is the Scherrer constant or shape factor ($K = 0.94$ for full width at half maximum measurements), $\Lambda$ is the wavelength of the X-ray, $B$ is the full width at half maximum of the fit profile to the peak at a given $2\theta$, and $\theta$ is the Bragg angle associated with the peak of interest. Detailed procedures for identifying the size, density, spacing, and orientation of crystalline domains are discussed in the Supplementary Text. Each sample is repeated at least three times with similar results.

## Molecular dynamics simulations

We use all-atom molecular dynamic (MD) simulations to model the thermal conductivity of mechanically strained PEG ideal networks. The COMPASS force field is used for these simulations. COMPASS is a class II force field parametrized for organic molecules, inorganic molecules, and polymers[63,64]. The COMPASS force field has been successfully used to study both thermal transport and mechanical properties in polymer systems due to its accurate parametrization of macroscopic properties, conformational energies, and molecular vibrations. The cut-off distance for pair interactions is set at 10 Å. Long-range electrostatic interactions are calculated using the particle-particle particle-mesh PPPM algorithm, and Lennard-Jones tail corrections are included. Following the original parametrization of the COMPASS force field for PEG units, a background dielectric constant of 1.4 is used[64]. Supplementary Fig. S18 illustrates the detailed setup procedures for sample initialization steps, stretching and equilibration steps, and production steps. More details on the molecular dynamics simulations are in the Supplementary Text.

## Reporting summary

Further information on research design is available in the Nature Portfolio Reporting Summary linked to this article.

# Data availability

The data generated in this study—including thermal conductivity measurements, X-ray characterizations, and molecular dynamics simulations—are provided in the Main Text and the Supplementary Information. Additionally, the original and processed datasets on X-ray scattering and thermal conductivity have been made publicly accessible via the public repository figshare (https://figshare.com) and are available here: https://doi.org/10.6084/m9.figshare.25287391. Data relevant to this study are available from the authors upon request.

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

## Acknowledgements

This work is supported by the U.S. Army Research Office through the Institute for Soldier Nanotechnologies at MIT (W911NF-13-D-0001) (X.Z.), National Institutes of Health (Grants No. 1R01HL153857-01 and No. 1R01HL167947-01) (X.Z.), National Science Foundation (Grant No. EFMA-1935291) (X.Z.), Department of Defense Congressionally Directed Medical Research Programs (Grant No. PR200524P1) (X.Z.), and Department of Energy (DE-FG02-02ER45977) (S.B.). C.H. acknowledges the support of the NSF Graduate Research Fellowship, the Warren M. Rohsenow Fellowship, the Epp and Ain Sonin Fellowship, and the MathWorks Fellowship. B.L. acknowledges the support of the Evergreen Graduate Innovation Fellowship. J.H.Z. acknowledges the support of the IBUILD Graduate Research Fellowship and the Abdul Latif Jameel Water and Food Systems Lab Graduate Fellowship. S.L. and Y.Z. acknowledge the startup fund from the College of Engineering at Michigan State University. The authors acknowledge Dr. Jordan Cox and Dr. Charles Settens for help with X-ray diffraction. This work is supported in part by the Koch Institute Support (core) Grant P30-CA14051 from the National Cancer Institute. The authors thank the Koch Institute's Robert A. Swanson (1969) Biotechnology Center for technical support, specifically Peterson (1957) Nanotechnology Materials Core Facility (RRID:SCR_018674). The authors acknowledge Dr. Margaret Bisher and Dr. David Mankus for helping with surface polishing via microtome cutting. The authors acknowledge Mrs. Jiabin Liu for the relaxation photoelasticimetry experiments. The authors acknowledge the MIT SuperCloud and Lincoln Laboratory Supercomputing Center for providing HPC resources that have contributed to the atomic simulation results.

## Author contributions

X.Z. and S.L. conceived the idea. X.Z. and G.C. supervised the research. S.L. and C.H. prepared the PEG samples. C.H. performed the small- and wide-angle X-ray scattering measurements. C.H. and S.L. prepared the samples for thermal conductivity measurements. B.L. performed the steady-state and FDTR thermal conductivity measurements. Z.Y. performed the photoelasticimetry experiments to quantify the structural relaxation time. J.Z. carried out atomic simulations. S.L., C.H., B.L., J.Z., Z.Y., G.L., J.S., S.B., G.C., and X.Z. analyzed and interpreted the results. S.L. drafted the manuscript with input from all other authors.

## Competing interests

The authors declare no competing interests.
