## [Peer Review File · Nature Communications]

Reversible Two-way Tuning of Thermal Conductivity in an End-linked Star-shaped ThermosetREVIEWER COMMENTS

Reviewer #1 (Remarks to the Author):

The manuscript authored by Shaoting Lin et al. presents an exploration of a polymer-based thermal switch, utilizing an end-linked star-shaped thermoset (ELST) to modulate thermal conductivity through both strain and temperature variations. The study reports an on/off ratio of up to 11.5 and estimates a response time of 6.4 seconds. While this research explores a pertinent topic within the field of thermal management, several critical issues must be addressed for a comprehensive evaluation:

1. A pivotal concern revolves around the lack of experimental validation for the claimed rapid response time of 6.4 seconds. Given the emphasized importance of this feature, it is imperative that the study incorporates in-situ and transient thermal conductivity measurements to substantiate this claim.
2. The thermal switch's response time, influenced by both mechanical and thermal modulation, raises concerns about inherent slowness. This slowness can be attributed to factors like sluggish phonon diffusion and the gradual changes in these two external fields, especially when compared to electric and optical stimuli. The authors should elucidate, with greater detail, how the thermal switch, reliant on relatively slower external fields, achieves the purported faster response time, as depicted in Figure 2e. Addressing challenges in achieving a swift response is essential.
3. The manuscript attributes ELST's switchable thermal conductivity to synergistic mechanisms involving aligned amorphous chains, oriented crystalline domains, and increased crystallinity induced by both strain and temperature fields. Given that these external fields often overlap, as temperature changes are concurrent with strain variations (e.g., thermal expansion), it is advisable to decouple and analyze these mechanisms for each control field. A more detailed analysis is necessary to isolate and study these mechanisms individually, providing a clearer understanding of the switch's performance under each external stimulus.
4. To contextualize the findings, the authors should compare the performance of the thermal switch with a recently published study in *Science* (*Science* 382, 585–589, 2023), involving an electrically gated molecular thermal switch with an on/off ratio of 15 and ultrahigh switching speeds of 1 MHz (response time of μs). This comparison will offer insights into the relative strengths and limitations of the proposed thermal switch.
5. The study employs two distinct experimental schemes (a steady-state system and a frequency domain thermoreflectance method, FDTR) to measure the thermal conductivity of ELST. To enhance clarity, the authors should provide insights into the testing accuracy of each scheme and elucidate the methods employed for its determination. Notably, in Figs.S3b and c, the presence of apparent dust or impurities on the equipment raises concerns about potential influences on the measurements. It would be valuable for the authors to detail the measures undertaken to mitigate the impact of such contaminants and ensure the elimination of contact thermal resistance between the sample and the experimental setup.
6. The switch's operation relying on two external fields raises concerns about practical feasibility. A comprehensive discussion on the practicality of integrating this thermal switch

into real-world scenarios, along with potential challenges and proposed solutions for system implementation, would be immensely valuable.

In summary, the manuscript shows promise but requires substantial revision. In its current form, I cannot recommend its publication in Nature Communications unless the noted issues have been carefully addressed.

Reviewer #2 (Remarks to the Author):

In this manuscript, the authors reported a polymeric thermal switch with a large on/off ratio of 11.5 and a fast response time of 6 s. These performances were attributed to the structural evolution of the polymer under the temperature and strain fields. This manuscript shows some interesting results, however, my biggest concern is how do the authors individually distinguish the multi-effects on the thermal conductivity, such as the multi-external fields (temperature and strain fields) and the ordering of lattice structures (aligned chains and oriented domains). I would suggest the authors to decouple the multi-mechanism effects so that the readers can have a clearer understanding on the story. Another problem is that I did not find the raw data about the response time. I would suggest the authors to include such data to prove the claimed high-speed of switching-response. The above questions should be carefully addressed before considering a publication.

Reviewer #3 (Remarks to the Author):

This manuscript presents a method for the reversible and rapid tuning of thermal conductivity in an end-linked, star-shaped polymer. Tuning can be achieved through both strain and temperature control. The results are intriguing and compelling, and I recommend publication after addressing the following points:

1. In Figure 2e, it would be beneficial to indicate whether the thermal switching is reversible, as the authors deem this an important parameter for evaluating switch performance.
2. In Figure 3c, the peaks at a scattering angle below 25° in the WAXS intensity profile show little change with stretch, while the peaks above 25° demonstrate a clear variation. Is this variation responsible for the claimed changes in crystallinity upon stretching, as shown in Figure 3f?
3. The experiments and molecular dynamics simulations use different molecular weights (20,000 and 10,000) of the ELST. Could this discrepancy potentially compromise the analysis of the mechanism?
4. There appears to be a noticeable change in thermal conductivity at a high stretch ratio ($\lambda = 10$) in cyclic stretch tests. What could be the reason for this, and could it potentially compromise the thermal switching function of the ELST?
5. What is the thickness of the transducer layer in the FDTR measurements? Is there a dependence of the measured thermal conductivity on the modulation frequency? The authors should provide more details about these measurements.

Response to Review Comments (Manuscript ID: NCOMMS-23-42501-T)
**“Reversible Two-way Tuning of Thermal Conductivity
in an End-linked Star-shaped Thermoset”**

Response to Reviewer 1

General comment. The manuscript authored by Shaoting Lin et al. presents an exploration of a polymer-based thermal switch, utilizing an end-linked star-shaped thermoset (ELST) to modulate thermal conductivity through both strain and temperature variations. The study reports an on/off ratio of up to 11.5 and estimates a response time of 6.4 seconds. While this research explores a pertinent topic within the field of thermal management, several critical issues must be addressed for a comprehensive evaluation. In summary, the manuscript shows promise but requires substantial revision. In its current form, I cannot recommend its publication in Nature Communications unless the noted issues have been carefully addressed.

Response to general comment. Thank you for acknowledging the significance of our work. We appreciate your valuable comments and suggestions, which provide insights that further strengthen the paper. In the following sections, we address each comment point by point. Newly inserted text in the manuscript and supplementary information are marked in **red**.

Comment 1. A pivotal concern revolves around the lack of experimental validation for the claimed rapid response time of 6.4 seconds. Given the emphasized importance of this feature, it is imperative that the study incorporates in-situ and transient thermal conductivity measurements to substantiate this claim.

Response to comment 1. Thank you for your insightful comments and suggestions. We fully recognize the advantages of incorporating in-situ thermal conductivity measurements as direct evidence to substantiate the claim of rapid response time. However, we must acknowledge that conducting in-situ thermal conductivity measurements using either the steady-state (SS) method or the frequency domain thermoreflectance (FDTR) method is technically unfeasible, due to the following reasons.

Technical barriers faced by the SS method for in-situ thermal conductivity measurements.

- *Requirement of sufficient time to establish an equilibrium state.* The measurement of thermal conductivity using the SS method requires a significant amount of time to ensure the establishment of an equilibrium thermal state. Such long measuring time guarantees stabilization of the sample, heater, and cooler within the surrounding environment, which are crucial for precise thermal flux measurement under controlled strain and temperature gradients. This equilibrium period typically extends to at least tens of minutes, which is much longer than the duration required for stretching or heating the sample. This therefore renders the in-situ thermal conductivity measurement impossible.
- *Requirements for sample preparation.* The in-situ thermal conductivity measurements using the SS method remains problematic because the hot and cool ends of the measurement apparatus must be detached and then reattached before and after stretching, which requires manual adjustment and opening of the thermal chamber. This process takes minutes followed

by an additional cycle of thermal equilibration. Furthermore, an in-situ stretching mechanism could easily damage the equipment, which contains fragile components in a chamber with limited space.

Technical barriers faced by the FDTR method for in-situ thermal conductivity measurements.

- *Requirement of sample embedding.* Thermal conductivity measurement using the FDTR method requires the sample to be embedded in epoxy for its surface to be exposed, followed by surface cutting and polishing. The procedure of sample embedding makes the in-situ thermal conductivity measurement impossible. First, the epoxy mold resists additional sample stretching by fixing the sample in place. The sample would need to be removed and then cast again between measurements. Second, the sample and epoxy interfaces are temperature sensitive; temperature changes cause nonuniform swelling that impacts the quality of the embedding and exposed surface.
- *Requirement of demanding surface quality.* To achieve a precise thermal conductivity measurement with the FDTR method, a high-quality surface is critical. Any strain or temperature fluctuation experienced by the sample alters its surface conditions, significantly impacting the measured thermal conductivity. As a result, the sample needs to be recut and repolished for an accurate reading. The procedures of recutting and repolishing render the in-situ thermal conductivity impossible.

While there are reported methods for in-situ thermal conductivity measurements of polymers subjected to changes in water content (e.g., J. A. Tomko, et al., *Nat. Nanotechnol.*, 13, 959-964, 2018), these methods are not applicable to polymers undergoing mechanical deformation. To the best of our knowledge, there are currently no documented techniques capable of achieving in-situ thermal conductivity measurements for polymers undergoing variations in strain and temperature.

Figure R1. Design of photoelasticimetry experiments for in-situ structural characterizations. **a**, Schematic illustration of the test setup for photoelasticimetry experiments, which contains one light source, one camera, two linear polarizers, and two quarter wave plates. A crack is introduced at the center of the sample to amplify its stress levels for enhancing light intensity. **b**, Representative measured fringe patterns in a stressed polymer.

Although direct in-situ thermal conductivity measurements are unfeasible, we conducted new photoelasticimetry experiments for in-situ structural characterizations, aiming to quantify the response time for structural alternation in the sample. Since the polymer's thermal conductivity is inherently linked to its molecular structure, the response time for structural changes in the sample

dictates its response time for thermal conductivity tuning. Notably, the wide range of thermal conductivity tuning (e.g., $\kappa_{\text{on}}/\kappa_{\text{off}} = 11.5$, 81% of the widest thermal conductivity tuning ratio reported in this work) can be achieved solely through strain modulation (**Fig. 2a** in the Main Text) with temperature variation serving as an additional design feature (**Fig. 2b** in the Main Text). Therefore, our focus in the photoelasticimetry experiments was on in-situ structural characterizations in the sample subjected to strain. As depicted in **Fig. R1a**, we designed a photoelasticimetry experimental setup to measure the internal stress of the polymer based on the

Figure R2. Comparison photoelasticimetry experiments. **a**, Schematic illustration of an instantaneous stretch applied on the sample followed by stress relaxation. **b**, Image sequences of fringe patterns in the PAAm subjected to an instantaneous stretch λ of 1.8 along the vertical direction followed by stress relaxation for up to 1000s. **c**, Image sequences of fringe patterns in our ELST subjected to an instantaneous stretch λ of 6.3 along the horizontal direction followed by stress relaxation for up to 200s. **d**, Light intensity at the crack tip of the PAAm versus time, identifying its response time for structural alternation of around 100 seconds. **e**, Light intensity at the crack tip of our ELST versus time, identifying its response time for structural alternation of around 3 seconds.

observed changes in light intensity resulting from alternations in its molecule structure. To enhance light intensity and amplify stress levels, we introduced a crack at the center of the sample to induce stress concentration. When the sample undergoes mechanical loading, a visual pattern of fringes (**Fig. R1b**), referred to as a photoelastic response, becomes apparent. This fringe pattern correlates with the internal stress associated with the molecular structure. When the sample is subjected to an instantaneous load, the fringe pattern typically undergoes changes over time, eventually stabilizing into a steady-state pattern as time approaches a critical value (**Fig. R2b**). The critical time scale for achieving the steady-state pattern defines the response time for structural alternations in the sample. The response time for structural alternations theoretically aligns with the response time for thermal conductivity tuning since the polymer's thermal conductivity is inherently linked to its molecular structure.

We specifically performed comparison photoelasticimetry experiments on the end-linked star-shaped thermoset (ELST) and a polyacrylamide-glycerol hydrogel (PAAm) representing a conventional polymer and demonstrated a substantially reduced response time for structural alternation in the ELST. As illustrated in **Fig. R2a**, both the ELST and PAAm samples are subjected to an instantaneous stretch (i.e., stretch of 6.3 in 1 second for the ELST, and stretch of 1.8 in 1 second for the PAAm). As shown in **Fig. R2b** and **Fig. R2d**, the response time for structural alternation of the PAAm sample is around 100 seconds; in contrast, as shown in **Fig. R2c** and **2e**, the response time for structural alternation of the ELST sample is only 3 seconds, orders of magnitude shorter than that of the PAAm. The short response time for structural alternation in the ELST serves as indirect evidence to substantiate the claim of rapid thermal conductivity tuning in the ELST.

We further performed mechanical characterizations to show that the ultra-short response time for structural alternation in the ELST is attributed to its low topological defects and negligible molecular entanglements. The low topological defects of the ELST are manifested by the

Figure R3. Mechanical characterizations. **a**, Cyclic nominal stress versus stretch curve of ELST, indicating its low stress-stretch hysteresis due to low molecular defects. **b**, Dynamic mechanical analysis of ELST, indicating its much lower loss tangent $\tan \delta$ compared to common polymers such as natural rubber due to negligible molecular entanglements.

negligible stress-stretch hysteresis, $h = \oint_1^{\lambda_{max}} s d\lambda$, where λ_{max} refers to the maximum stretch of the ELST subjected to one cycle of loading and unloading (**Fig. R3a**) and negligible molecular entanglements as demonstrated by the much-reduced loss tangent $\tan \delta$, which is an order of magnitude lower than natural rubber (**Fig. R3b**). More detailed interpretations can be referred to in our recent paper (Hartquist et al., *Sci. Adv.*, 9, 50, 2023).

Revision for comment 1. In the Main Text and Supplementary Material, we make three major changes to incorporate these revisions. First, we adjust the wording to tune down the claim of rapid response time for thermal conductivity tuning. Second, we include the supplementary stress-relaxation photoelasticimetry experiments to serve as indirect evidence substantiating the rapid response time in thermal conductivity tuning. Third, while we tune down the claim of rapid response time for thermal conductivity tuning, we highlight the unique features of the ELST, particularly its reversible nature and significant thermal conductivity tuning capability, which surpass that of existing polymeric thermal switches.

Specifically, we remove “rapid” in the title, changing to “**Reversible Two-way Tuning of Thermal Conductivity in an End-linked Star-shaped Thermoset**”.

In lines 216 to 237, we add “**We further estimate the response time for the two-way tuning of thermal conductivity in the ELST, a critical characteristic of thermal switches. The total response time of the two-way tuning of thermal conductivity includes two parts: the time for mechanical or thermal modulation and the response time for structural relaxation. Since the ELST’s thermal conductivity is primarily affected by strain (Fig. 2a, c), we particularly focus on the time for mechanical modulation, which is necessary for large thermal conductivity tuning ratios up to 11.5. The time for mechanical modulation is determined by the time for stretching the material, which can be estimated by $\tau_{\text{strain}} = (\lambda - 1)L/V$, where λ is the stretch ratio, L is the sample’s gage length, and V is the loading speed. Given $\lambda = 20$, $L = 2$ mm, and $V = 5$ mm/s, the time for stretching the material is estimated as $\tau_{\text{strain}} = 7.2$ s. The time for thermal modulation is mainly dominated by the time for thermal conduction, which can be estimated by $\tau_{\text{heat}} = t^2/D_{\text{heat}}$, where t is the sample thickness and D_{heat} is the thermal diffusivity of the sample. Given $t \sim 1$ mm and $D_{\text{heat}} = 0.36$ mm²/s⁴⁶, the time for thermal conduction is estimated as $\tau_{\text{heat}} = 2.8$ s. We further performed photoelasticimetry experiments (Supplementary Fig. S9)^{47,48} to quantify the response time for structural alternation in the ELST subjected to a nearly instantaneous stretch (i.e., stretch of 6.3 in 1 s). As shown in Supplementary Fig. S10c, e, the response time for structural alternation of the ELST sample is only 3 s, orders of magnitude shorter than that of conventional polymers (e.g., 100 s in polyacrylamide hydrogel shown in Supplementary Fig. S10b, d). The short response time for structural relaxation in the ELST is attributed to its unique low topological defects and negligible molecular entanglements as demonstrated in our recent paper³⁹. Considering both the time scale of thermal and mechanical stimuli, and the structural response time, we estimate that the switching time to be ~ 10 s”.** In the Supplementary Material, we add **Section 4** entitled Photoelasticimetry Experiments, **Supplementary Fig. S9**, and **Supplementary Fig. S10**.

To highlight the unique features of the ELST, in line 214-215, we add “**Notably, the reversible and significant thermal conductivity tuning capabilities of our ELST have not been achieved in existing**

polymeric thermal switches”. We particularly compare our ELST to existing reversible polymeric thermal switches. Our ELST demonstrates a substantially improved tuning ratio and notably shortened tuning time (**Fig. R4a**), while preserving these features on a bulk scale (**Fig. R4b**). In line 243-246, we add “Particularly, when compared to existing reversible polymeric thermal switches^{30, 31, 54}, our ELST demonstrates a substantially improved tuning ratio and notably shortened tuning time (**Fig. 3b**), while preserving these features on a bulk scale (**Fig. 3c** and **Supplementary Table 1**).”

Figure R4. Comparison of thermal conductivity tuning performances between the ELST and existing reversible polymeric thermal switches. a, Comparison chart in the plot of thermal conductivity on/off tuning ratio κ_{on}/κ_{off} versus response time τ and **b**, Comparison chart in the plot of thermal conductivity on/off tuning ratio κ_{on}/κ_{off} versus characteristic length l comparing our ELST to existing reversible polymeric thermal switches.

Comment 2. The thermal switch’s response time, influenced by both mechanical and thermal modulation, raises concerns about inherent slowness. This slowness can be attributed to factors like sluggish phonon diffusion and the gradual changes in these two external fields, especially when compared to electric and optical stimuli. The authors should elucidate, with greater detail, how the thermal switch, reliant on relatively slower external fields, achieves the purported faster response time, as depicted in Figure 2e. Addressing challenges in achieving a swift response is essential.

Response to comment 2. Many thanks for the reviewer’s comments on the thermal switch’s response time. While we agree that thermal modulation induces a response time required for

phonon diffusion and mechanical modulation involves a response time required for stress relaxation, we respectfully disagree that thermal and mechanical modulations suffer inherent slowness compared to other stimuli, such as electrical, chemical, and magnetic fields. Although these external stimuli can be applied faster, the change in thermal conductivity is determined by the time-constant of the structural changes caused by these stimuli, which could be long. For example, thermal switches relying on optical stimuli require a response time associated with structural changes in response to optical stimuli, taking tens of seconds (Shin et al., *PNAS*, 116, 13, 2019). Thermal switches relying on magnetic fields require a response time associated with structural changes in response to magnetic field, taking tens of minutes (Shin et al., *ACS Macro Letters*, 5, 955-960, 2016). Thermal switches based on electrochemical intercalation require a response time associated with the diffusion and reaction of species, spanning from tens of minutes (Sood et al., *Nat. Commun.*, 9, 4510, 2018) to hours (Cho et al., *Nat. Commun.*, 5, 4035, 2014).

In contrast, the response time for thermal and mechanical modulations can be shorter than ten seconds depending on the sample geometry. Using the ELST studied in this work as one example, the response time associated with phonon diffusion due to thermal modulation can be estimated by $\tau_{\text{heat}} \approx t^2/D_{\text{heat}} \approx 2.8$ s, where $t \sim 1$ mm is the sample thickness in this work and $D_{\text{heat}} = 0.36$ mm²/s is the thermal diffusivity of the PEG, sharing the same chemistry as the ELST. The response time associated with stress relaxation due to mechanical modulation can be estimated by the summation of the time for mechanical stretching τ_{stretch} and the time for structural relaxation $\tau_{\text{structural}}$. The time for mechanical stretching can be estimated by $\tau_{\text{stretch}} \approx (\lambda - 1)L/V \approx 7.2$ s, where $\lambda \approx 20$ is the stretch ratio applied on the sample, $L = 2$ mm is the sample's gage length, and $V = 5$ mm/s is the loading speed applied on the sample. The time for structural relaxation $\tau_{\text{structural}}$ is measured as 3 s using our new photoelasticity experiments (**Fig. R2c** and **e**). Note that the time for structural relaxation in conventional polymers typically spans from 100 s to 1000 s. Our ELST exhibits an ultra-short response time for structural relaxation, attributed to its unique features of low topological defects and negligible molecular entanglements (Hartquist et al., *Sci. Adv.*, 9, 50, 2023). Thus, the total response time associated with stress relaxation due to mechanical modulation is around 10.2 s.

We agree that there are thermal switches based on electrical fields exhibiting ultra-fast response times on the order of seconds (Foley et al., *ACS Appl. Mater. Interfaces*, 10, 25493–25501, 2018; Ihlefeld et al., *Nano Lett.*, 15, 1791-1795, 2015). However, these thermal switches often exhibit a low thermal conductivity tuning ratio below 3. In addition to the response time for thermal conductivity tuning, reversibility and tuning ratio are also critical characteristics of thermal switches. Notably, the reversible and significant thermal conductivity tuning capabilities of our ELST have not been achieved in existing polymeric thermal switches. Our updated **Fig. 3** and **Table S1** summarize the combined properties of thermal conductivity tuning of our ELST, including its reversibility, tuning ratio, and response time, which collectively position our ELST as superior to most existing thermal switches.

It should be noted that the recently reported electrically gated solid-state thermal switch exhibits an ultra-short response time up to 10^{-6} s while maintaining an ultra-high thermal conductivity tuning ratio up to 13 (Li et al, *Science*, 382, 585-589, 2023), significantly pushing the limit of existing thermal switches. **However, we note that such switching is not polymeric and limited to**

monolayer interfacial region. In contrast, our polymeric ELST can be fabricated on a bulk scale through mass production.

Table S1. Comparison between the ELST and existing thermal switches

Materials	Mechanism	Polymer	Tuning ratio	Response time (s)	Continuous	Reversibility	Size	Cycles	Ref
SrCoO _{2.5}	Electrochemical intercalation	No	1.5	10800	Yes	Yes	27-44 nm thickness	2	[20]
MoS ₂	Electrochemical intercalation	No	10	900	Yes	Yes	~ 10 nm thickness	2	[21]
Graphene nanoparticle	Electrical field	No	1.4	1	Yes	Yes	/	4	[22]
VO ₂ doped with tungsten	Metal-insulator transition	No	1.5	/	No	/	~ 10 μm length ~ 100 nm diameter	/	[23]
Graphite/hexadecane	Solid-liquid transition	No	3.2	/	No	/	/	/	[24]
CNT/hexadecane	Solid-liquid transition	No	3.0	/	No	/	/	/	[25]
Lead zirconate titanate (PZT)	Domain polarization by electrical field	No	1.1	20	No	Yes	~ 100 nm thickness	5	[26]
Bismuth-antimonide alloy	Domain polarization by magnetic field	No	1.2	/	No	/	1-20 μm thickness	/	[27]
Carboranethiol cage molecules	Self-assembled molecular junctions	No	13	10 ⁻⁶	Yes	Yes	Monolayer	10 ⁶	[28]
Graphene composite foam	Domain transformation by pressure	No	8.0	1000	Yes	Yes	1.2 mm thickness	2	[29]
Azobenzene polymer	Chain configuration by light	Yes	3.5	100	No	Yes	280 nm thickness	6	[30]
Liquid crystal polymer	Chain configuration by magnetic field	Yes	1.4	600	Yes	Yes	/	/	[31]
Tandem-repeat protein	Chain configuration by hydration	Yes	4.0	140	No	Yes	~ 100 μm thickness	1	[32]
PE nanofiber	Chain configuration by temperature	Yes	8.0	/	No	/	105 nm diameter	/	[33]
PE film	Chain configuration by thermal drawing	Yes	163	/	No	No	150 μm thickness ~ 10 cm length	/	[4]
ELST	Chain configuration by strain and temperature modulation	Yes	11.5	9.4	Yes	Yes	~ 1 mm thickness ~ 1 cm length	1000	This work

Revision for comment 2. We made the following changes in the Main Text. In lines 216 to 250, we rewrote the paragraph “We further estimate the response time for the two-way tuning of thermal conductivity in the ELST, a critical characteristic of thermal switches. The total response time of the two-way tuning of thermal conductivity includes two parts: the time for mechanical or thermal modulation and the response time for structural relaxation. Since the ELST’s thermal conductivity is primarily affected by strain (Fig. 2a, c), we particularly focus on the time for mechanical modulation, which is necessary for large thermal conductivity tuning ratios up to 11.5. The time for mechanical modulation is determined by the time for stretching the material, which can be estimated by $\tau_{\text{strain}} = (\lambda - 1)L/V$, where λ is the stretch ratio, L is the sample’s gage length, and V is the loading speed. Given $\lambda = 20$, $L = 2$ mm, and $V = 5$ mm/s, the time for stretching the material is estimated as $\tau_{\text{strain}} = 7.2$ s. The time for thermal modulation is mainly dominated by the time for thermal conduction, which can be estimated by $\tau_{\text{heat}} = t^2/D_{\text{heat}}$, where t is the sample thickness and D_{heat} is the thermal diffusivity of the sample. Given $t \sim 1$ mm and $D_{\text{heat}} = 0.36$ mm²/s⁴⁶, the time for thermal conduction is estimated as $\tau_{\text{heat}} = 2.8$ s. We further performed photoelasticity experiments (Supplementary Fig. S9)^{47,48} to quantify the response time for structural alternation in the ELST subjected to a nearly instantaneous stretch (i.e., stretch

of 6.3 in 1 s). As shown in **Supplementary Fig. S10c, e**, the response time for structural alternation of the ELST sample is only 3 s, orders of magnitude shorter than that of conventional polymers (e.g., 100 s in polyacrylamide hydrogel shown in **Supplementary Fig. S10b, d**). The short response time for structural relaxation in the ELST is attributed to its unique low topological defects and negligible molecular entanglements as demonstrated in our recent paper³⁹. Considering both the time scale of thermal and mechanical stimuli, and the structural response

Fig. 3 | Thermal conductivity tuning performances of the 20,000 MW ELST. a, Comparison chart in the plot of thermal conductivity on/off tuning ratio κ_{on}/κ_{off} versus response time τ , comparing our ELST to existing reversible thermal switches such as an electrically gated molecular thermal switch⁵⁵, electrochemical layered materials^{7, 8, 49}, nanoparticle suspensions^{50, 51}, ferroelectric materials^{14, 52}, and deformable composites^{3, 53}. **b**, Comparison chart in the plot of thermal conductivity on/off tuning ratio κ_{on}/κ_{off} versus response time τ , comparing our ELST to existing reversible polymeric thermal switches^{30, 31, 54}. **c**, Comparison chart in the plot of thermal conductivity on/off tuning ratio κ_{on}/κ_{off} versus characteristic length l , comparing our ELST to existing reversible polymeric thermal switches^{30, 31, 54}.

time, we estimate that the switching time to be ~ 10 s. **Figure 3a** summarizes the comparison chart in the plot of thermal conductivity on/off tuning ratio $\kappa_{\text{on}}/\kappa_{\text{off}}$ versus response time τ . Our ELST exhibits a thermal conductivity on/off tuning ratio up to 11.5 at a fixed temperature of $T = 60$ °C, up to 2.3 at a fixed stretch of 2.5, up to 14.2 through a combined strain and temperature modulation, and response time on the order of 10 s, outperforming most existing thermal switches such as electrochemical layered materials^{7, 8, 49}, nanoparticle suspensions^{50, 51}, ferroelectric materials^{14, 52}, and deformable composites^{3, 53}. Particularly, when compared to existing reversible polymeric thermal switches^{30, 31, 54}, our ELST demonstrates a substantially improved tuning ratio and notably shortened tuning time (**Fig. 3b**), while preserving these features on a bulk scale (**Fig. 3c** and **Supplementary Table 1**). It should be noted that the recently reported electrically gated solid-state thermal switch⁵⁵ exhibits an ultra-short response time up to 10^{-6} s while maintaining an ultra-high thermal conductivity tuning ratio up to 13, significantly pushing the limit of existing thermal switches including our ELST. However, such fast modulation so far is on a monolayer at the interface region. In contrast, our ELST can be fabricated on a bulk scale through mass production.”

In addition, we replot **Fig. 3a**, which includes the recently reported electrically gated solid-state thermal switch. To highlight the unique features of the ELST other than response time that surpass existing polymeric thermal switches, we add in lines 214-215 “Notably, the reversible and significant thermal conductivity tuning capabilities of our ELST have not been achieved in existing polymeric thermal switches”. We also add **Fig. 3b, c** to compare thermal conductivity tuning performance of our ELST to existing reversible polymeric thermal switches.

Comment 3. The manuscript attributes ELST’s switchable thermal conductivity to synergistic mechanisms involving aligned amorphous chains, oriented crystalline domains, and increased crystallinity induced by both strain and temperature fields. Given that these external fields often overlap, as temperature changes are concurrent with strain variations (e.g., thermal expansion), it is advisable to decouple and analyze these mechanisms for each control field. A more detailed analysis is necessary to isolate and study these mechanisms individually, providing a clearer understanding of the switch's performance under each external stimulus.

Response to comment 3. Many thanks for reviewer’s great comments. We fully agree the importance of decoupling the effects from strain and temperature fields. To decouple the effects of strain and temperature, we replot thermal conductivity of the ELST as a function of stretch ratio at $T = 30$ °C and 60 °C and thermal conductivity of the ELST as a function of temperature at $\lambda = 1$ and 20. As shown in **Fig. R5a**, thermal conductivity of the ELST increases with the stretch ratio, indicating the crucial role of mechanical strain in tuning thermal conductivity. We further study the effect of temperature on thermal conductivity tuning. As shown in **Fig. R5b**, the rise in temperature slightly decreases thermal conductivity of the ELST at the undeformed state (i.e., $\lambda = 1$) due to the melting of randomly oriented crystalline domains. These figures suggest that the dominant effect is mechanical while thermal modulation becomes more pronounced at higher stretching ratios, providing additional tuning capacity when used in conjunction.

Figure R5. Effects of strain and temperature on thermal conductivity of ELST. a, Thermal conductivity of the ELST as a function of stretch ratio at $T = 30$ and $60\text{ }^{\circ}\text{C}$. **b,** Thermal conductivity of the ELST as a function of temperature at $\lambda = 1$ and 20 .

To further decouple the two-way tuning mechanisms, we performed two series of x-ray scattering characterizations to investigate the strain (**Fig. R6**) and temperature (**Fig. R7**) effects individually.

Figure R6. The strain effect on ELST structure characterized via X-ray scattering. a, SAXS and WAXS scattering patterns and **b,** intensity profiles of the ELST at different stretch ratios at $T = 25\text{ }^{\circ}\text{C}$. **c,** SAXS and WAXS scattering patterns and **d,** intensity profiles of the ELST at different stretch ratios at $T = 55\text{ }^{\circ}\text{C}$. **e,** Crystallinity, **f,** Herman's orientation factor, **g,** Start-to-start spacing L , and **h,** crystalline domain size D versus stretch ratio λ at $T = 25$ and $55\text{ }^{\circ}\text{C}$.

First, to investigate the strain effect, we performed X-ray scattering characterizations on the ELST subjected to various stretch levels at a low temperature below the ELST's melting point (i.e., $T = 25\text{ }^{\circ}\text{C}$) and at a high temperature above ELST's melting point (i.e., $T = 55\text{ }^{\circ}\text{C}$). As shown in **Fig. R6**, at $T = 25\text{ }^{\circ}\text{C}$, the ELST exhibits a slight increase of crystallinity (**Fig. R6c**), a highly enhanced Herman's orientation factor at moderate deformations (**Fig. R6d**), a gradual increase of start-to-start spacing L between two crystallites (**Fig. R6e**), and a nearly constant crystalline size D (**Fig. R6f**) as the stretch ratio increases. In contrast, at $T = 55\text{ }^{\circ}\text{C}$, the ELST exhibits a significantly increased crystallinity with increasing stretching ratio (**Fig. R6c**), a nearly constant Herman's orientation factor at large deformation (**Fig. R6d**), a slight increase of start-to-start spacing L at large deformations (**Fig. R6e**), and a nearly constant crystalline size D at large deformations (**Fig. R6f**).

Second, to investigate the temperature effect, we also performed X-ray scattering characterizations on the ELST under different temperatures at the undeformed state (i.e., $\lambda = 1$) and at the highly deformed state (i.e., $\lambda = 20$). As shown in **Fig. R7**, at $\lambda = 1$, the ELST exhibits a significant decrease of crystallinity above $45\text{ }^{\circ}\text{C}$ (**Fig. R7c**), a nearly constant Herman's orientation factor (**Fig. R7d**), a gradual increase of start-to-start spacing L (**Fig. R7e**), and a nearly constant crystalline size D (**Fig. R7f**) as temperature increases. The evolution of these structural parameters implies the slight reduction of thermal conductivity at an elevated temperature in the undeformed ELST (**Fig. R5b**)

Figure R7. The temperature effect on ELST structure characterized via X-ray scattering. **a**, SAXS and WAXS scattering patterns and **b**, intensity profiles of ELST exposed to different temperatures at undeformed state (i.e., $\lambda = 1$). **c**, SAXS and WAXS scattering patterns and **d**, intensity profiles of ELST exposed to different temperatures at highly deformed state (i.e., $\lambda = 20$). **e**, Crystallinity, **f**, Herman's orientation factor, **g**, Start-to-start spacing L , and **h**, crystalline domain size D versus temperature T of ELST at undeformed state (i.e., $\lambda = 1$) and at highly deformed state (i.e., $\lambda = 20$).

is strictly attributed to the decreased density of crystalline domains and increase in randomly oriented amorphous chains. In contrast, at $\lambda = 20$, the ELST exhibits a significantly decreased crystallinity above 45 °C (Fig. R7c), a constant Herman's orientation factor (Fig. R7d), an increase of start-to-start spacing L at high temperature (Fig. R7e), and a nearly constant crystalline size D (Fig. R7f). The evolution of these structural parameters indicates two effects due to temperature rise: 1) one minor contribution from reduced strain-induced crystallization, and 2) one dominant contribution from increased oriented amorphous chains between crystalline domains. These two effects synergistically modulate the ELST's thermal conductivity.

Revision for comment 3. In the Main Text, we implemented two major modifications. First, we reorganized the measured thermal conductivity of the ELST, investigating the effects of strain and temperature individually. Second, we performed X-ray scattering characterizations of the ELST to investigate the separate impacts of strain and temperature on the two-way thermal transport mechanism in the ELST.

Specifically, in line 159-181, we revised the main text to “We next measure the thermal conductivity of a series of tetra-arm PEG thermosets with various stretch ratios and at different temperatures. We first investigate the impact of strain on thermal conductivity tuning. As shown in Fig. 2a, the undeformed ELST is found to have in-plane thermal conductivity of $0.24 \text{ W m}^{-1} \text{ K}^{-1}$ at $T = 30 \text{ °C}$ (Supplementary Fig. S4). As the stretch ratio increases, the thermal conductivity of the deformed tetra-PEG thermosets along the stretch direction significantly increases, reaching $1.42 \text{ W m}^{-1} \text{ K}^{-1}$ at $T = 30 \text{ °C}$. Notably, the ELST exhibits a maximum strain-modulated thermal conductivity enhancement up to 11.5 from 0.15 to $2.1 \text{ W m}^{-1} \text{ K}^{-1}$ at $T = 60 \text{ °C}$ (Fig. 2c). We further study the effect of temperature on thermal conductivity tuning. As shown in Fig. 2b, the rise in temperature slightly decreases thermal conductivity of the ELST at undeformed state (i.e., $\lambda = 1$). The maximum temperature-modulated thermal conductivity tuning ratio is 2.3 when the ELST is subjected to a fixed stretch of 2.5 (Fig. 2d). Notably, as shown in Supplementary Fig. S4, at a fixed small stretch ratio (i.e., $\lambda = 2.5, 5$), the temperature shows negligible impacts on the on/off thermal conductivity tuning ratio; in contrast, at a fixed large stretch ratio (i.e., $\lambda = 10, 15, 20$), the temperature increase significantly enhances the on/off thermal conductivity tuning ratio $\kappa_{\text{on}}/\kappa_{\text{off}}$ up to 11.5. We interpret the slight drop in thermal conductivity with increasing temperature at small deformations is due to the melting of unoriented crystalline domains. Although significant change in the crystallinity occurs above 45 °C, we do not observe a large change of thermal conductivity because the thermal conductivity of polymers is governed predominately by the amorphous regime¹⁷. The nuanced variation in thermal conductivity in the deformed ELST at an elevated temperature is potentially a result of two competing phenomena: 1) the hinderance of phonon transport in the ELST due to the crystalline-to-amorphous transition, and 2) the augmentation of phonon transport in the ELST due to the aligned polymer chains.”

In line 278-335, we added a new paragraph to discuss the decoupled strain and temperature effect in the two-way thermal transport tuning mechanism. “We use X-ray characterization to investigate strain as the first mechanism for thermal conductivity tuning (Fig. 4). ELSTs are stretched and cooled to room temperature and fixed on both ends with Crazy glue to an acrylic mount (See details in Supplementary Information). Figure 4a, b plot the small-angle X-ray scattering (SAXS) and wide-angle X-ray scattering (WAXS) patterns of the samples under different stretch

ratios at room temperature (25 °C), respectively. We first employ the measured WAXS intensity profile to quantify the crystallinity in the undeformed ELST (**Fig. 4c**). As shown in **Supplementary Fig. S13c**, the presence of narrow peaks denotes the formation of crystalline domains. The crystallinity of the sample can be quantified by fitting the measured WAXS intensity distributions with Gaussian or Pseudo-Voigt functions (**Supplementary Fig. S15**). As shown in **Fig. 4c**, the measured crystallinity of the undeformed nearly ideal-network polymer at $T = 25$ °C is measured to be 48%. As the stretch ratio increases, the crystallinity of the deformed ELST gradually increases up to 66%, indicating a pronounced increase of crystallinity as stretch ratio increases. Next, we analyze the azimuthal spread of diffraction peak intensity to characterize the orientation of crystalline domains. The orientation order of crystalline domains is assessed using Hermans' orientation factor, defined as $f_2 = \frac{3\langle \cos^2\phi \rangle - 1}{2}$, where ϕ is the azimuthal angle (**Supplementary Fig. S16**). As shown in **Fig. 4d**, the undeformed ELST shows nearly zero Hermans' orientation factor, suggesting the nature of randomly distributed crystalline domains. As the stretch ratio reaches 5, the Hermans' orientation factor significantly increases to 0.9, validating that oriented domains dominate in the tetra-PEG thermoset at moderate deformations. As the stretch ratio increases, the Hermans' orientation factor further increases up to 1.0, suggesting that crystalline domains become almost fully oriented along the stretch direction. Given the measured WAXS and SAXS data, we further quantify the size of crystalline domains D and start-to-start spacing L between adjacent crystalline domains for the ELST at various stretch ratios. As summarized in **Fig. 4f**, the size of crystalline domains remains constant at around 12 nm, while the distance between repeating structural units of crystalline and amorphous domains gradually increases from 8 nm to 17 nm as the stretch ratio increases (**Fig. 4e**). The preservation of crystalline domains justifies the consistent crystalline domain size as the ELST's are heated, stretched, and cooled; furthermore, the increased distance between crystalline domains indicates the alignment of amorphous chains in between crystalline domains. At $T = 55$ °C above T_m , we observe an intriguing strain-induced crystallinity up to 50% when stretched to 20 (**Fig. 4b, c**), significantly outperforming natural rubber (i.e., 14% at 22 °C and 10% at 55 °C) and other well-studied elastomers including butyl rubber, polyisoprene rubber, and polybutadiene rubber³⁹. In addition, we find that the temperature rise can further increase the Hermans' orientation factor of the ELST, indicating a further orientation of crystalline domains at $T = 55$ °C (**Fig. 4d**). Furthermore, since the polymer chain is more flexible at $T = 55$ °C, the start-to-start distance spacing L between adjacent crystalline domains increases from 20 nm to 23 nm as the stretch ratio increases, larger than that at $T = 25$ °C (**Fig. 4e**). Notably, the size of crystalline domains D of the ELST at $T = 55$ °C remains almost the same as that at $T = 25$ °C (**Fig. 4f**). The ultra-high strain-induced crystallinity, the further increased orientation of crystalline domains, and the enlarged start-to-start distance spacing between adjacent crystalline domains in the ELST possibly explains the highly enhanced thermal conductivity tuning ratio up to 11.5 at an elevated temperature. We further use X-ray characterizations to study temperature modulation as the second thermal conductivity tuning mechanism (**Fig. 5**). **Figure 5a** plots the SAXS and WAXS scattering patterns of ELST under different temperatures at undeformed state ($\lambda = 1$). As shown in **Fig. 5c**, the crystallinity of ELST decreases with rising temperature, attributed to the melting of crystalline domains. Despite the significant change in the crystallinity occurs above T_m , the ELST's thermal conductivity exhibits

a slight drop because the thermal conductivity of polymers is governed predominately by the amorphous regime¹⁷. The undeformed ELST maintains an almost zero Hermans' orientation factor irrespective of temperature, indicating that temperature does not promote the orientation of crystalline domains. Furthermore, the start-to-start spacing L between adjacent crystalline domains increases as the temperature increases, which is due to the melting of crystalline domains that make crystalline domains sparser (**Fig. 5e**). Additionally, **Figure 5f** reveals no temperature dependency on crystalline size. Similarly, when ELST is highly deformed (i.e., $\lambda = 20$), the ELST shows a decrease in crystallinity (**Fig. 5c**), an augmentation in start-to-start spacing (**Fig. 5e**), a constant Hermans' orientation factor, and an unaltered crystalline size as the temperature increases.”

In addition, we reorganized **Figure 2, 3, 4** in the Main Text to present the individual effects of strain and temperature on thermal transport mechanism in ELST. We also added four supplementary figures (i.e., **Supplementary Figure 11-14**) and supplementary tables (i.e., **Supplementary Table 1,2**) to decouple the effects of strain and temperature.

Comment 4. To contextualize the findings, the authors should compare the performance of the thermal switch with a recently published study in *Science* (*Science* 382, 585–589, 2023), involving an electrically gated molecular thermal switch with an on/off ratio of 15 and ultrahigh switching speeds of 1 MHz (response time of μs). This comparison will offer insights into the relative strengths and limitations of the proposed thermal switch.

Response to comment 4. Many thanks to the reviewer for bringing us the attention of the recently published *Science* paper. The *Science* paper presents an electrically gated thermal switch utilizing solid-state molecular structures that demonstrate switching speeds exceeding 1 megahertz and on/off ratios in thermal conductance exceeding 13. The thermal conductivity switching speed of the electrically gated thermal switch surpasses that of our ELST. However, one potential drawback of the electrically gated thermal switch lies in its scalability. In contrast, our ELST offers the advantage of scalability and can be fabricated on a bulk scale through mass production.

Revision for comment 4. We made the following changes in the Main Text. We replot **Fig. 2e** as **Fig. 3a** in the current version, which includes the recently reported electrically gated solid-state thermal switch. In addition, in line 237 to 250, we made modifications as “**Figure 3a** summarizes the comparison chart in the plot of thermal conductivity on/off tuning ratio $\kappa_{\text{on}}/\kappa_{\text{off}}$ versus response time τ . Our ELST exhibits a thermal conductivity on/off tuning ratio up to 11.5 at a fixed temperature of $T = 60\text{ }^\circ\text{C}$, up to 2.3 at a fixed stretch of 2.5, up to 14.2 through a combined strain and temperature modulation, and response time on the order of 10 s, outperforming most existing thermal switches such as electrochemical layered materials^{7, 8, 49}, nanoparticle suspensions^{50, 51}, ferroelectric materials^{14, 52}, and deformable composites^{3, 53}. Particularly, when compared to existing reversible polymeric thermal switches^{30, 31, 54}, our ELST demonstrates a substantially improved tuning ratio and notably shortened tuning time (**Fig. 3b**), while preserving these features on a bulk scale (**Fig. 3c** and **Supplementary Table 1**). It should be noted that the recently reported electrically gated solid-state thermal switch⁵⁵ exhibits an ultra-short response time up to 10^{-6} s while maintaining an ultra-high thermal conductivity tuning ratio up to 13, significantly pushing the limit of existing thermal switches including our ELST. However, such fast modulation so far

is on a monolayer at the interface region. In contrast, our ELST can be fabricated on a bulk scale through mass production”.

Comment 5. The study employs two distinct experimental schemes (a steady-state system and a frequency domain thermoreflectance method, FDTR) to measure the thermal conductivity of ELST. To enhance clarity, the authors should provide insights into the testing accuracy of each scheme and elucidate the methods employed for its determination. Notably, in Fig. S3b and c, the presence of apparent dust or impurities on the equipment raises concerns about potential influences on the measurements. It would be valuable for the authors to detail the measures undertaken to mitigate the impact of such contaminants and ensure the elimination of contact thermal resistance between the sample and the experimental setup.

Response to comment 5. Many thanks for the reviewer’s valuable comments. We employed the steady-state method as the primary tool for thermal conductivity measurement while adopting the FDTR method as a validation technique. The measured thermal conductivity from the FDTR method matches trends and outcomes observed from the steady-state method. The testing accuracies and their determinations for both methods are reported in relevant references (Xu et al., *Nat. Commun.*, 10, 1771, 2019; Schmidt et al., *Rev. Sci. Instrum.*, 84, 10, 2013). For measurements reported in this work, the sample thermal conductivities varying from 0.18 to 2.10 W/mK are within the accuracies of both methodologies. For each of our reported values, an error bar is provided to justify the sensitivity (each measured value is much greater than its error bar).

For the steady state method, the systematic errors come from: 1. Deviation from the one-dimensional heat conduction assumption; 2. Heat loss to the environment through convection, radiation, and conduction. In the provided analysis, however, we have justified that both sources are either minimized or will only reflect a larger true value than reported (i.e., we provided conservative estimation of sample thermal conductivity). Measurement errors come from the readings of heater power, thermocouple measurements, and sample geometries. For each reported data point, at least three independent measurements of the same sample are performed. Error bars reflect the obtained standard deviations and are considered as accurate.

For the FDTR method, the systematic errors come from the deviation from the bilayer isotropic transient heat conduction model. The measurement errors come from the reading of pump spot radius, transducer layer thickness, and readings of reflectance signal intensity. We have carefully minimized the measurement errors by calibrating each batch of sputtering coated sample with a standard silicon wafer, with which we will get the spot radius and transducer thickness. The error bar is obtained through a Monte Carlo fitting, where 1000 iterations of curve fitting will be performed with randomized trial values of sample thermal conductivity and interfacial conductivity. A confidence interval is obtained during this Monte Carlo fitting process (Yang et al., *Rev. Sci. Instrum.*, 87, 1, 2016) which is then reported as error bars.

We also appreciate the careful inspection of the testing setup for additional sources of error. In **Supplementary Fig. S3b, c**, the “impurities” on our film are a silver thermal paste. This common thermal interfacial material is applied between the clamps and the sample to minimize the thermal interfacial resistance. In our model, we assume the thermocouples inserted in the hot and cold clamps directly measure the sample temperatures; therefore, this mechanism is necessary to

minimize the thermal interfacial resistance. Consequently, slight amounts of excess thermal paste can flow out as we fasten the clamp, as shown in **Supplementary Fig. S3b, c**. The squeezed-out paste cannot be easily cleaned without changing samples. We contend that the error introduced by having residual thermal paste is negligible because the length of “contaminated” sample area is much smaller than the total sample length, see **Supplementary Fig. S3b, c**. Effectively, this thin layer of residual thermal paste becomes a thermal resistance connected in parallel with our sample.

Considering the most conservative extreme, where the silver paste has infinite thermal conductivity, then its presence should effectively shorten the sample length between clamps from the measured value. Taking **Supplementary Fig. S3b** as an example (the sample length between clamps is 7 mm; the length of sample containing silver paste is 0.5 mm), the corrected sample thermal conductivity in the conservative extreme is 7.1% smaller than reported value.

Revision for comment 5. We have carefully checked our images and raw data to make sure all the reported values are not affected by the occasionally appearing residual thermal pastes.

Comment 6. The switch’s operation relying on two external fields raises concerns about practical feasibility. A comprehensive discussion on the practicality of integrating this thermal switch into real-world scenarios, along with potential challenges and proposed solutions for system implementation, would be immensely valuable.

Response to comment 6. Many thanks for the review comments on practical feasibility. First, it’s important to clarify that the substantial thermal conductivity tuning of the ELST can occur strictly via strain modulation, with temperature modulation serving as a secondary feature in modulating the ELST’s thermal conductivity. The ELST reaches a tuning ratio of 11.5 by strain alone; temperature change can be employed to elevate that ratio to 14 or preserve a stress-free deformed state through the shape memory effect. Second, our ELST finds significant relevance in scenarios involving large deformation. A notable application is in the development of a solid-state elastocaloric cooling system. Unlike many caloric materials, caloric polymers such as natural rubber and our ELST (Hartquist et al., *Sci. Adv.*, 9, 50, 2023), as demonstrated in our recent work, are soft, cost-effective, and environmentally friendly. The widespread availability of caloric polymers, coupled with the low mechanical forces required for stretching, positions them as ideal candidates for future large-scale, high-power cooling devices. One major limitation faced by existing caloric polymers is their inability to transfer heat to specific regions due to restricted thermal conductivity tuning of polymers. Our ELST’s two-way tuning of thermal conductivity expands the engineering possibilities, which allows us to push the boundaries of elastocaloric cooling performance while minimizing dissipation to the surrounding environment.

Revision for comment 6. In the Main Text, we add new discussions on the practicality of integrating our ELST in real-world applications. In line 403 to 406, we add “**For example, our ELST’s two-way tuning of thermal conductivity potentially expands engineering spaces for developing next generation of elastocaloric cooling materials, which allows us to push the boundaries of elastocaloric cooling performance while minimizing dissipation to the surrounding environment**”.

Response to Reviewer 2

Comment 1. In this manuscript, the authors reported a polymeric thermal switch with a large on/off ratio of 11.5 and a fast response time of 6 s. These performances were attributed to the structural evolution of the polymer under the temperature and strain fields. This manuscript shows some interesting results, however, my biggest concern is how do the authors individually distinguish the multi-effects on the thermal conductivity, such as the multi-external fields (temperature and strain fields) and the ordering of lattice structures (aligned chains and oriented domains). I would suggest the authors decouple the multi-mechanism effects so that the readers can have a clearer understanding on the story.

Response to comment 1. Many thanks for reviewer's great comments. We fully agree the importance of decoupling the effects from strain and temperature fields. To decouple the effects of strain and temperature, we replot thermal conductivity of the ELST as a function of stretch ratio at $T = 30\text{ }^{\circ}\text{C}$ and $60\text{ }^{\circ}\text{C}$ and thermal conductivity of the ELST as a function of temperature at $\lambda = 1$ and 20 . As shown in **Fig. R5a**, thermal conductivity of the ELST increases with the stretch ratio, indicating the crucial role of mechanical strain in tuning thermal conductivity. We further study the effect of temperature on thermal conductivity tuning. As shown in **Fig. R5b**, the rise in temperature slightly decreases thermal conductivity of the ELST at the undeformed state (i.e., $\lambda = 1$) due to the melting of randomly oriented crystalline domains. These figures suggest that the dominant effect is mechanical while thermal modulation becomes more pronounced at higher stretching ratios, providing additional tuning capacity when used in conjunction.

To further decouple the two-way tuning mechanisms, we performed two series of x-ray scattering characterizations to investigate the strain (**Fig. R6**) and temperature (**Fig. R7**) effects individually.

First, to investigate the strain effect, we performed X-ray scattering characterizations on the ELST subjected to various stretch levels at a low temperature below the ELST's melting point (i.e., $T =$

Figure R5. Effects of strain and temperature on thermal conductivity of ELST. a, Thermal conductivity of the ELST as a function of stretch ratio at $T = 30$ and $60\text{ }^{\circ}\text{C}$. **b,** Thermal conductivity of the ELST as a function of temperature at $\lambda = 1$ and 20 .

25 °C) and at a high temperature above ELST's melting point (i.e., $T = 55$ °C). As shown in **Fig. R6**, at $T = 25$ °C, the ELST exhibits a slight increase of crystallinity (**Fig. R6c**), a highly enhanced Herman's orientation factor at moderate deformations (**Fig. R6d**), a gradual increase of start-to-start spacing L between two crystallites (**Fig. R6e**), and a nearly constant crystalline size D (**Fig. R6f**) as the stretch ratio increases. In contrast, at $T = 55$ °C, the ELST exhibits a significantly increased crystallinity with increasing stretching ratio (**Fig. R6c**), a nearly constant Herman's orientation factor at large deformation (**Fig. R6d**), a slight increase of start-to-start spacing L at large deformations (**Fig. R6e**), and a nearly constant crystalline size D at large deformations (**Fig. R6f**).

Figure R6. The strain effect on ELST structure characterized via X-ray scattering. a, SAXS and WAXS scattering patterns and **b**, intensity profiles of the ELST at different stretch ratios at $T = 25$ °C. **c**, SAXS and WAXS scattering patterns and **d**, intensity profiles of the ELST at different stretch ratios at $T = 55$ °C. **e**, Crystallinity, **f**, Herman's orientation factor, **g**, Start-to-start spacing L , and **h**, crystalline domain size D versus stretch ratio λ at $T = 25$ and 55 °C.

Second, to investigate the temperature effect, we also performed X-ray scattering characterizations on the ELST under different temperatures at the undeformed state (i.e., $\lambda = 1$) and at the highly deformed state (i.e., $\lambda = 20$). As shown in **Fig. R7**, at $\lambda = 1$, the ELST exhibits a significant decrease of crystallinity above 45 °C (**Fig. R7c**), a nearly constant Herman's orientation factor (**Fig. R7d**), a gradual increase of start-to-start spacing L (**Fig. R7e**), and a nearly constant crystalline size D (**Fig. R7f**) as temperature increases. The evolution of these structural parameters implies the slight reduction of thermal conductivity at an elevated temperature in the undeformed ELST (**Fig. R5b**) is strictly attributed to the decreased density of crystalline domains and increase in randomly oriented amorphous chains. In contrast, at $\lambda = 20$, the ELST exhibits a significantly decreased

Figure R7. The temperature effect on ELST structure characterized via X-ray scattering. a, SAXS and WAXS scattering patterns and **b**, intensity profiles of ELST exposed to different temperatures at undeformed state (i.e., $\lambda = 1$). **c**, SAXS and WAXS scattering patterns and **d**, intensity profiles of ELST exposed to different temperatures at highly deformed state (i.e., $\lambda = 20$). **e**, Crystallinity, **f**, Herman's orientation factor, **g**, Start-to-start spacing L , and **h**, crystalline domain size D versus temperature T of ELST at undeformed state (i.e., $\lambda = 1$) and at highly deformed state (i.e., $\lambda = 20$).

crystallinity above 45 °C (**Fig. R7c**), a constant Herman's orientation factor (**Fig. R7d**), an increase of start-to-start spacing L at high temperature (**Fig. R7e**), and a nearly constant crystalline size D (**Fig. R7f**). The evolution of these structural parameters indicates two effects due to temperature rise: 1) one minor contribution from reduced strain-induced crystallization, and 2) one dominant contribution from increased oriented amorphous chains between crystalline domains. These two effects synergistically modulate the ELST's thermal conductivity.

Revision for comment 1. In the Main Text, we implemented two major modifications. First, we reorganized the measured thermal conductivity of the ELST, investigating the effects of strain and temperature individually. Second, we performed X-ray scattering characterizations of the ELST to investigate the separate impacts of strain and temperature on the two-way thermal transport mechanism in the ELST.

Specifically, in line 159-181, we revised the main text to “We next measure the thermal conductivity of a series of tetra-arm PEG thermosets with various stretch ratios and at different temperatures. We first investigate the impact of strain on thermal conductivity tuning. As shown in **Fig. 2a**, the undeformed ELST is found to have in-plane thermal conductivity of $0.24 \text{ W m}^{-1} \text{ K}^{-1}$ at $T = 30 \text{ }^\circ\text{C}$ (**Supplementary Fig. S4**). As the stretch ratio increases, the thermal conductivity of the deformed tetra-PEG thermosets along the stretch direction significantly increases, reaching $1.42 \text{ W m}^{-1} \text{ K}^{-1}$ at $T = 30 \text{ }^\circ\text{C}$. Notably, the ESLT exhibits a maximum strain-modulated thermal

conductivity enhancement up to 11.5 from 0.15 to 2.1 W m⁻¹ K⁻¹ at $T = 60$ °C (**Fig. 2c**). We further study the effect of temperature on thermal conductivity tuning. As shown in **Fig. 2b**, the rise in temperature slightly decreases thermal conductivity of the ELST at undeformed state (i.e., $\lambda = 1$). The maximum temperature-modulated thermal conductivity tuning ratio is 2.3 when the ELST is subjected to a fixed stretch of 2.5 (**Fig. 2d**). Notably, as shown in **Supplementary Fig. S4**, at a fixed small stretch ratio (i.e., $\lambda = 2.5, 5$), the temperature shows negligible impacts on the on/off thermal conductivity tuning ratio; in contrast, at a fixed large stretch ratio (i.e., $\lambda = 10, 15, 20$), the temperature increase significantly enhances the on/off thermal conductivity tuning ratio $\kappa_{\text{on}}/\kappa_{\text{off}}$ up to 11.5. We interpret the slight drop in thermal conductivity with increasing temperature at small deformations is due to the melting of unoriented crystalline domains. Although significant change in the crystallinity occurs above 45 °C, we do not observe a large change of thermal conductivity because the thermal conductivity of polymers is governed predominately by the amorphous regime¹⁷. The nuanced variation in thermal conductivity in the deformed ELST at an elevated temperature is potentially a result of two competing phenomena: 1) the hinderance of phonon transport in the ELST due to the crystalline-to-amorphous transition, and 2) the augmentation of phonon transport in the ELST due to the aligned polymer chains.”

In line 278-335, we added a new paragraph to discuss the decoupled strain and temperature effects in the two-way thermal transport tuning mechanism. “We use X-ray characterization to investigate strain as the first mechanism for thermal conductivity tuning (**Fig. 4**). ELSTs are stretched and cooled to room temperature and fixed on both ends with Krazy glue to an acrylic mount (See details in **Supplementary Information**). **Figure 4a,b** plot the small-angle X-ray scattering (SAXS) and wide-angle X-ray scattering (WAXS) patterns of the samples under different stretch ratios at room temperature (25 °C), respectively. We first employ the measured WAXS intensity profile to quantify the crystallinity in the undeformed ELST (**Fig. 4c**). As shown in **Supplementary Fig. S13c**, the presence of narrow peaks denotes the formation of crystalline domains. The crystallinity of the sample can be quantified by fitting the measured WAXS intensity distributions with Gaussian or Pseudo-Voight functions (**Supplementary Fig. S15**). As shown in **Fig. 4c**, the measured crystallinity of the undeformed nearly ideal-network polymer at $T = 25$ °C is measured to be 48%. As the stretch ratio increases, the crystallinity of the deformed ELST gradually increases up to 66%, indicating a pronounced increase of crystallinity as stretch ratio increases. Next, we analyze the azimuthal spread of diffraction peak intensity to characterize the orientation of crystalline domains. The orientation order of crystalline domains is assessed using Hermans’ orientation factor, defined as $f_2 = \frac{3\langle \cos^2\phi \rangle - 1}{2}$, where ϕ is the azimuthal angle

(**Supplementary Fig. S16**). As shown in **Fig. 4d**, the undeformed ELST shows nearly zero Hermans’ orientation factor, suggesting the nature of randomly distributed crystalline domains. As the stretch ratio reaches 5, the Hermans’ orientation factor significantly increases to 0.9, validating that oriented domains dominate in the tetra-PEG thermoset at moderate deformations. As the stretch ratio increases, the Hermans’ orientation factor further increases up to 1.0, suggesting that crystalline domains become almost fully oriented along the stretch direction. Given the measured WAXS and SAXS data, we further quantify the size of crystalline domains D and start-to-start spacing L between adjacent crystalline domains for the ELST at various stretch ratios. As summarized in **Fig. 4f**, the size of crystalline domains remains constant at around 12 nm, while

the distance between repeating structural units of crystalline and amorphous domains gradually increases from 8 nm to 17 nm as the stretch ratio increases (**Fig. 4e**). The preservation of crystalline domains justifies the consistent crystalline domain size as the ELST's are heated, stretched, and cooled; furthermore, the increased distance between crystalline domains indicates the alignment of amorphous chains in between crystalline domains. At $T = 55\text{ }^{\circ}\text{C}$ above T_m , we observe an intriguing strain-induced crystallinity up to 50% when stretched to 20 (**Fig. 4b, c**), significantly outperforming natural rubber (i.e., 14% at 22 $^{\circ}\text{C}$ and 10% at 55 $^{\circ}\text{C}$) and other well-studied elastomers including butyl rubber, polyisoprene rubber, and polybutadiene rubber³⁹. In addition, we find that the temperature rise can further increase the Hermans' orientation factor of the ELST, indicating a further orientation of crystalline domains at $T = 55\text{ }^{\circ}\text{C}$ (**Fig. 4d**). Furthermore, since the polymer chain is more flexible at $T = 55\text{ }^{\circ}\text{C}$, the start-to-start distance spacing L between adjacent crystalline domains increases from 20 nm to 23 nm as the stretch ratio increases, larger than that at $T = 25\text{ }^{\circ}\text{C}$ (**Fig. 4e**). Notably, the size of crystalline domains D of the ELST at $T = 55\text{ }^{\circ}\text{C}$ remains almost the same as that at $T = 25\text{ }^{\circ}\text{C}$ (**Fig. 4f**). The ultra-high strain-induced crystallinity, the further increased orientation of crystalline domains, and the enlarged start-to-start distance spacing between adjacent crystalline domains in the ELST possibly explains the highly enhanced thermal conductivity tuning ratio up to 11.5 at an elevated temperature. We further use X-ray characterizations to study temperature modulation as the second thermal conductivity tuning mechanism (**Fig. 5**). **Figure 5a** plots the SAXS and WAXS scattering patterns of ELST under different temperatures at undeformed state ($\lambda = 1$). As shown in **Fig. 5c**, the crystallinity of ELST decreases with rising temperature, attributed to the melting of crystalline domains. Despite the significant change in the crystallinity occurs above T_m , the ELST's thermal conductivity exhibits a slight drop because the thermal conductivity of polymers is governed predominately by the amorphous regime¹⁷. The undeformed ELST maintains an almost zero Hermans' orientation factor irrespective of temperature, indicating that temperature does not promote the orientation of crystalline domains. Furthermore, the start-to-start spacing L between adjacent crystalline domains increases as the temperature increases, which is due to the melting of crystalline domains that make crystalline domains sparser (**Fig. 5e**). Additionally, **Figure 5f** reveals no temperature dependency on crystalline size. Similarly, when ELST is highly deformed (i.e., $\lambda = 20$), the ELST shows a decrease in crystallinity (**Fig. 5c**), an augmentation in start-to-start spacing (**Fig. 5e**), a constant Hermans' orientation factor, and an unaltered crystalline size as the temperature increases.”

In addition, we reorganized **Figure 2, 3, 4** in the Main Text to present the individual effects of strain and temperature on thermal transport mechanism in ELST. We also added four supplementary figures (i.e., **Supplementary Figure 11-14**) and supplementary tables (i.e., **Supplementary Table 1-2**) to decouple the effects of strain and temperature.

Comment 2. Another problem is that I did not find the raw data about the response time. I would suggest the authors include such data to prove the claimed high-speed of switching-response. The above questions should be carefully addressed before considering a publication.

Response to comment 2. Thank you for your insightful comments and suggestions. We fully recognize the advantages of incorporating in-situ thermal conductivity measurements as direct evidence to substantiate the claim of rapid response time. However, we must acknowledge that

conducting in-situ thermal conductivity measurements using either the steady-state (SS) method or the frequency domain thermoreflectance (FDTR) method is technically unfeasible, due to the following reasons.

Technical barriers faced by the SS method for in-situ thermal conductivity measurements.

- *Requirement of sufficient time to establish an equilibrium state.* The measurement of thermal conductivity using the SS method requires a significant amount of time to ensure the establishment of an equilibrium thermal state. Such long measuring time guarantees stabilization of the sample, heater, and cooler within the surrounding environment, which are crucial for precise thermal flux measurement under controlled strain and temperature gradients. This equilibrium period typically extends to at least tens of minutes, which is much longer than the duration required for stretching or heating the sample. This therefore renders the in-situ thermal conductivity measurement impossible.
- *Requirements for sample preparation.* The in-situ thermal conductivity measurements using the SS method remains problematic because the hot and cool ends of the measurement apparatus must be detached and then reattached before and after stretching, which requires manual adjustment and opening of the thermal chamber. This process takes minutes followed by an additional cycle of thermal equilibration. Furthermore, an in-situ stretching mechanism could easily damage the equipment, which contains fragile components in a chamber with limited space.

Technical barriers faced by the FDTR method for in-situ thermal conductivity measurements.

- *Requirement of sample embedding.* Thermal conductivity measurement using the FDTR method requires the sample to be embedded in epoxy for its surface to be exposed, followed by surface cutting and polishing. The procedure of sample embedding makes the in-situ thermal conductivity measurement impossible. First, the epoxy mold resists additional sample stretching by fixing the sample in place. The sample would need to be removed and then cast again between measurements. Second, the sample and epoxy interfaces are temperature sensitive; temperature changes cause nonuniform swelling that impacts the quality of the embedding and exposed surface.
- *Requirement of demanding surface quality.* To achieve a precise thermal conductivity measurement with the FDTR method, a high-quality surface is critical. Any strain or temperature fluctuation experienced by the sample alters its surface conditions, significantly impacting the measured thermal conductivity. As a result, the sample needs to be recut and repolished for an accurate reading. The procedures of recutting and repolishing render the in-situ thermal conductivity impossible.

While there are reported methods for in-situ thermal conductivity measurements of polymers subjected to changes in water content (e.g., J. A. Tomko, et al., *Nat. Nanotechnol.*, 13, 959-964, 2018), these methods are not applicable to polymers undergoing mechanical deformation. To the best of our knowledge, there are currently no documented techniques capable of achieving in-situ thermal conductivity measurements for polymers undergoing variations in strain and temperature.

Figure R1. Design of photoelasticimetry experiments for in-situ structural characterizations. **a**, Schematic illustration of the test setup for photoelasticimetry experiments, which contains one light source, one camera, two linear polarizers, and two quarter wave plates. A crack is introduced at the center of the sample to amplify its stress levels for enhancing light intensity. **b**, Representative measured fringe patterns in a stressed polymer.

Although direct in-situ thermal conductivity measurements are unfeasible, we conducted new photoelasticimetry experiments for in-situ structural characterizations, aiming to quantify the response time for structural alternation in the sample. Since the polymer's thermal conductivity is inherently linked to its molecular structure, the response time for structural changes in the sample dictates its response time for thermal conductivity tuning. Notably, the wide range of thermal conductivity tuning (e.g., $\kappa_{\text{on}}/\kappa_{\text{off}} = 11.5$, 81% of the widest thermal conductivity tuning ratio reported in this work) can be achieved solely through strain modulation (**Fig. 2a** in the Main Text) with temperature variation serving as an additional design feature (**Fig. 2b** in the Main Text). Therefore, our focus in the photoelasticimetry experiments was on in-situ structural characterizations in the sample subjected to strain. As depicted in **Fig. R1a**, we designed a photoelasticimetry experimental setup to measure the internal stress of the polymer based on the observed changes in light intensity resulting from alternations in its molecule structure. To enhance light intensity and amplify stress levels, we introduced a crack at the center of the sample to induce stress concentration. When the sample undergoes mechanical loading, a visual pattern of fringes (**Fig. R1b**), referred to as a photoelastic response, becomes apparent. This fringe pattern correlates with the internal stress associated with the molecular structure. When the sample is subjected to an instantaneous load, the fringe pattern typically undergoes changes over time, eventually stabilizing into a steady-state pattern as time approaches a critical value (**Fig. R2b**). The critical time scale for achieving the steady-state pattern defines the response time for structural alternations in the sample. The response time for structural alternations theoretically aligns with the response time for thermal conductivity tuning since the polymer's thermal conductivity is inherently linked to its molecular structure.

We specifically performed comparison photoelasticimetry experiments on the end-linked star-shaped thermoset (ELST) and a polyacrylamide-glycerol hydrogel (PAAm) representing a conventional polymer and demonstrated a substantially reduced response time for structural alternation in the ELST. As illustrated in **Fig. R2a**, both the ELST and PAAm samples are subjected to an instantaneous stretch (i.e., stretch of 6.3 in 1 second for the ELST, and stretch of 1.8 in 1 second for the PAAm). As shown in **Fig. R2b** and **Fig. R2d**, the response time for structural alternation of the PAAm sample is around 100 seconds; in contrast, as shown in **Fig. R2c** and **2e**,

Figure R2. Comparison photoelasticimetry experiments. **a**, Schematic illustration of an instantaneous stretch applied on the sample followed by stress relaxation. **b**, Image sequences of fringe patterns in the PAAm subjected to an instantaneous stretch λ of 1.8 along the vertical direction followed by stress relaxation for up to 1000s. **c**, Image sequences of fringe patterns in our ELST subjected to an instantaneous stretch λ of 6.3 along the horizontal direction followed by stress relaxation for up to 200s. **d**, Light intensity at the crack tip of the PAAm versus time, identifying its response time for structural alternation of around 100 seconds. **e**, Light intensity at the crack tip of our ELST versus time, identifying its response time for structural alternation of around 3 seconds.

the response time for structural alternation of the ELST sample is only 3 seconds, orders of magnitude shorter than that of the PAAm. The short response time for structural alternation in ELST serves as indirect evidence to substantiate the claim of rapid thermal conductivity tuning in the ELST.

We further performed mechanical characterizations to show that the ultra-short response time for structural alternation in the ELST is attributed to its low topological defects and negligible molecular entanglements. The low topological defects of the ELST are manifested by the

Figure R3. Mechanical characterizations. **a**, Cyclic nominal stress versus stretch curve of ELST, indicating its low stress-stretch hysteresis due to low molecular defects. **b**, Dynamic mechanical analysis of ELST, indicating its much lower loss tangent $\tan \delta$ compared to common polymers such as natural rubber due to negligible molecular entanglements.

negligible stress-stretch hysteresis, $h = \oint_1^{\lambda_{max}} s d\lambda$, where λ_{max} refers to the maximum stretch of the ELST subjected to one cycle of loading and unloading (**Fig. R3a**) and negligible molecular entanglements as demonstrated by the much-reduced loss tangent $\tan \delta$, which is an order of magnitude lower than natural rubber (**Fig. R3b**). More detailed interpretations can be referred to in our recent paper (Hartquist et al., *Sci. Adv.*, 9, 50, 2023).

Revision for comment 2. In the Main Text and Supplementary Material, we make three major changes to incorporate these revisions. First, we adjust the wording to tune down the claim of rapid response time for thermal conductivity tuning. Second, we include the supplementary stress-relaxation photoelasticimetry experiments to serve as indirect evidence substantiating the rapid response time in thermal conductivity tuning. Third, while we tune down the claim of rapid response time for thermal conductivity tuning, we highlight the unique features of the ELST, particularly its reversible nature and significant thermal conductivity tuning capability, which surpass that of existing polymeric thermal switches.

Specifically, we remove “rapid” in the title, changing to “**Reversible Two-way Tuning of Thermal Conductivity in an End-linked Star-shaped Thermoset**”.

In lines 216 to 237, we add “**We further estimate the response time for the two-way tuning of thermal conductivity in the ELST, a critical characteristic of thermal switches. The total response time of the two-way tuning of thermal conductivity includes two parts: the time for mechanical or thermal modulation and the response time for structural relaxation. Since the ELST’s thermal conductivity is primarily affected by strain (Fig. 2a, c), we particularly focus on the time for mechanical modulation, which is necessary for large thermal conductivity tuning ratios up to 11.5. The time for mechanical modulation is determined by the time for stretching the material, which can be estimated by $\tau_{strain} = (\lambda - 1)L/V$, where λ is the stretch ratio, L is the sample’s gage length, and V is the loading speed. Given $\lambda = 20$, $L = 2$ mm, and $V = 5$ mm/s, the time for**

stretching the material is estimated as $\tau_{\text{strain}} = 7.2$ s. The time for thermal modulation is mainly dominated by the time for thermal conduction, which can be estimated by $\tau_{\text{heat}} = t^2/D_{\text{heat}}$, where t is the sample thickness and D_{heat} is the thermal diffusivity of the sample. Given $t \sim 1$ mm and $D_{\text{heat}} = 0.36$ mm²/s⁴⁶, the time for thermal conduction is estimated as $\tau_{\text{heat}} = 2.8$ s. We further performed photoelasticimetry experiments (**Supplementary Fig. S9**)^{47,48} to quantify the response time for structural alternation in the ELST subjected to a nearly instantaneous stretch (i.e., stretch of 6.3 in 1 s). As shown in **Supplementary Fig. S10c, e**, the response time for structural alternation of the ELST sample is only 3 s, orders of magnitude shorter than that of conventional polymers (e.g., 100 s in polyacrylamide hydrogel shown in **Supplementary Fig. S10b, d**). The short response time for structural relaxation in the ELST is attributed to its unique low topological defects and negligible molecular entanglements as demonstrated in our recent paper³⁹. Considering both the time scale of thermal and mechanical stimuli, and the structural response time, we estimate that the switching time to be ~ 10 s". In the Supplementary Material, we add **Section 4** entitled Photoelasticimetry Experiments, **Supplementary Fig. S9**, and **Supplementary Fig. S10**.

To highlight the unique features of the ELST, in line 214-215, we add "Notably, the reversible and significant thermal conductivity tuning capabilities of our ELST have not been achieved in existing polymeric thermal switches". We particularly compare our ELST to existing reversible polymeric

Figure R4. Comparison of thermal conductivity tuning performances between the ELST and existing reversible polymeric thermal switches. a, Comparison chart in the plot of thermal conductivity on/off tuning ratio $\kappa_{\text{on}}/\kappa_{\text{off}}$ versus response time τ and **b**, Comparison chart in the plot of thermal conductivity on/off tuning ratio $\kappa_{\text{on}}/\kappa_{\text{off}}$ versus characteristic length l comparing our ELST to existing reversible polymeric thermal switches.

thermal switches. Our ELST demonstrates a substantially improved tuning ratio and notably shortened tuning time (**Fig. R4a**), while preserving these features on a bulk scale (**Fig. R4b**). In line 243-246, we add “**Particularly, when compared to existing reversible polymeric thermal switches** ^{30, 31, 54}, our ELST demonstrates a substantially improved tuning ratio and notably shortened tuning time (**Fig. 3b**), while preserving these features on a bulk scale (**Fig. 3c and Supplementary Table 1**).”

Response to Reviewer 3

General comment. This manuscript presents a method for the reversible and rapid tuning of thermal conductivity in an end-linked, star-shaped polymer. Tuning can be achieved through both strain and temperature control. The results are intriguing and compelling, and I recommend publication after addressing the following points.

Response to general comment. Many thanks to the reviewer for acknowledging the novelty and recommending the publication of this work. We appreciate your valuable comments and suggestions, which provide insights that further strengthen the paper. In the following sections, we address each comment point by point. Newly inserted text in the manuscript and supplementary information are marked in red.

Comment 1. In Figure 2e, it would be beneficial to indicate whether the thermal switching is reversible, as the authors deem this an important parameter for evaluating switch performance.

Response and revision for comment 1. Thanks for the comment. We agree that reversibility is an important parameter for evaluating switch performance. In the figure caption, we indicate the thermal switches summarized in the current version as **Fig. 3** are reversible. In line 583, “Comparison chart in the plot of thermal conductivity on/off tuning ratio $\kappa_{\text{on}}/\kappa_{\text{off}}$ versus response time τ , comparing our tetra-PEG thermosets to existing thermal switches existing reversible thermal switches ...”. In addition, we replot **Fig. 2e** in the new **Fig. 3** to compare the performance of our ELST to existing thermal switches and existing reversible polymeric thermal switches.

Comment 2. In Figure 3c, the peaks at a scattering angle below 25° in the WAXS intensity profile show little change with stretch, while the peaks above 25° demonstrate a clear variation. Is this variation responsible for the claimed changes in crystallinity upon stretching, as shown in Fig. 3f?

Response to comment 2. Many thanks for the comment. At $T = 25^\circ$ below T_m , since the ELST does not experience crystalline-to-amorphous transition, the WAXS intensity profiles at different stretches exhibit little variation. While the WAXS intensity profiles at different stretches show little difference, the presence of stretch gradually increases crystallinity (**Fig. 4c**), slightly increases Herman’s orientation factor (**Fig. 4d**), slightly increases start-to-start spacing (**Fig. 4e**) but has little impact on crystalline size (**Fig. 4f**). At $T = 55^\circ$ above T_m , the ELST experiences a significant strain-induced crystallization as demonstrated in our recent work (Hartquist et al., *Sci. Adv.*, 9, 50, 2023), the WAXS intensity profiles at different stretches show significant difference. Specifically, the presence of stretch significantly increases the crystallinity at large deformations (**Fig. 4c**), significantly increases the Herman’s orientation factor at moderate deformations (**Fig. 4d**), slightly increases start-to-start spacing (**Fig. 4e**) but has little impact on crystalline size (**Fig. 4f**).

Revision for comment 2. To clarify the impacts of strain at different temperatures and decouple the effects of strain and temperature in the two-way tuning thermal transport mechanism. We replot the schematic in **Supplementary Fig. S11** to illustrate the effects of strain and temperature: 1) The

applied mechanical strain promotes phonon transport due to the synergy of oriented crystalline domains, aligned interstitial amorphous chains, and increased crystallinity. 2) The increased temperature induces the hinderance of phonon transport due to the crystalline-to-amorphous transition and the augmentation of phonon transport due to the aligned polymer chains, given their increased flexibility. In addition, we reorganize **Figure 2-4** in the main text to present the individual effects of strain and temperature on thermal transport mechanism in the ELST. We also

Fig. 3 | Thermal conductivity tuning performances of the 20,000 MW ELST. a, Comparison chart in the plot of thermal conductivity on/off tuning ratio κ_{on}/κ_{off} versus response time τ , comparing our ELST to existing reversible thermal switches such as an electrically gated molecular thermal switch⁵⁵, electrochemical layered materials^{7, 8, 49}, nanoparticle suspensions^{50, 51}, ferroelectric materials^{14, 52}, and deformable composites^{3, 53}. **b**, Comparison chart in the plot of thermal conductivity on/off tuning ratio κ_{on}/κ_{off} versus response time τ , comparing our ELST to existing reversible polymeric thermal switches^{30, 31, 54}. **c**, Comparison chart in the plot of thermal conductivity on/off tuning ratio κ_{on}/κ_{off} versus characteristic length l , comparing our ELST to existing reversible polymeric thermal switches^{30, 31, 54}.

add four supplementary figures (i.e., **Supplementary Figure 11-14**) and two supplementary tables (i.e., **Supplementary Table 1-2**) to decouple the effects of strain and temperature.

Comment 3. The experiments and molecular dynamics simulations use different molecular weights (20,000 and 10,000) of the ELST. Could this discrepancy potentially compromise the analysis of the mechanism?

Response to comment 3. The lower 10k MW was chosen to make the simulations more computationally feasible due to the large number of atoms involved in simulating the ELST topology. **Figures 6b, d** highlight that polymer chain alignment along the stretching direction dominates in enabling the large increase in thermal conductivity. Due to the ELST's unique feature with low topological defects and negligible molecular entanglements, it is believed that the overall trends and results of the simulations will hold for larger MW ELST. The low topological defects of the ELST are manifested by the negligible stress-stretch hysteresis, $h = \oint_1^\lambda s d\lambda$, where λ refers to the maximum stretch of the ELST subjected to one cycle of loading and unloading (**Fig. R3a**) and negligible molecular entanglements as demonstrated by the much-reduced loss tangent $\tan \delta$, which is an order of magnitude lower than natural rubber (**Fig. R3b**). More detailed interpretations for both molecular weights can be referred to in our recent paper (Hartquist et al., *Sci. Adv.*, 9, 50, 2023). The primary difference between a 10k and 20k MW sample will be that the increase in maximum stretch ratio will also increase the final thermal conductivity value due to the longer polymer chains between cross-linkers. It is also noted that the highlighted polymer structure changes in simulations that lead to the increased thermal conductivity align well with the experimental conclusions of the 20k ELSTs.

Figure R3. Mechanical characterizations. **a**, Cyclic nominal stress versus stretch curve of ELST, indicating its low stress-stretch hysteresis due to low molecular defects. **b**, Dynamic mechanical analysis of ELST, indicating its much lower loss tangent $\tan \delta$ compared to common polymers such as natural rubber due to negligible molecular entanglements.

Comment 4. There appears to be a noticeable change in thermal conductivity at a high stretch ratio ($\lambda = 10$) in cyclic stretch tests. What could be the reason for this, and could it potentially compromise the thermal switching function of the ELST?

Response to comment 4. Many thanks for the reviewer’s very insightful observation. We did observe the noticeable change in thermal conductivity at a high stretch ratio in cyclic tests. The reason for this trend is due to the mechanical training that further increases crystallinity, orients crystalline domains, and aligns interstitial polymer chains, which was reported in a PVA hydrogel material that contains a similar semi-crystalline architecture (S Lin et al., *PNAS*, 116, 21, 2019). While the thermal conductivity of the ELST changes over cycle numbers at small cycles of loading, its thermal conductivity should reach a steady state value as manifested by its stress-stretch curves under cyclic loading (**Fig. R8**).

Revision for comment 4. We provide a brief interpretation for the noticeable change in thermal conductivity at a high stretch ratio in cyclic tests. In line 198-204, we add “**Notably, we observe a change in thermal conductivity as the cycle number increases, which is due to mechanical training that further increases crystallinity, orients crystalline domains, and aligns interstitial polymer chains, which matches results reported in a PVA hydrogel material with a similar semi-crystalline architecture. While the thermal conductivity of the ELST changes over cycle numbers at small cycles of loading, results suggest that the thermal conductivity should reach a steady state value as manifested by its stress-stretch curves under cyclic loading (Supplementary Fig. S8).**”

Figure R8. Maximum stress versus cycle number of the 20,000 MW ELST. Maximum stress reaches a steady state value as cycle number reaches around 20.

Comment 5. What is the thickness of the transducer layer in the FDTR measurements? Is there a dependence of the measured thermal conductivity on the modulation frequency? The authors should provide more details about these measurements.

Response to comment 5. We appreciate the reviewer’s comments on this point. The thickness of the gold transducer layer is 200 nm. There is an additional 10 nm binding layer of Titanium between the sample and transducer layer. FDTR is a measurement in the frequency domain, where a wide range of modulation frequencies are pumped, and the surface reflectance responses are

probed. In our measurement, we scan frequency range spanning from 2×10^3 Hz to 1×10^7 Hz. The phase lag and amplitude of the probed signal versus the pump frequency are fitted to get the polymer thermal conductivity and polymer-transducer interfacial thermal conductivity (**Fig. R9**). The reported thermal conductivity should be considered as the average thermal conductivity within the frequency range we measure. We have provided the measured phase and amplitude of our probe signal, as well as the probe response curve with the fitted thermal conductivity and interfacial conductance as a comparison. Two additional curves showing $\pm 10\%$ of the thermal conductivity are also provided to illustrate the sensitivity of the FDTR signal on sample properties.

It should be noted that we employed the steady-state method as the primary tool for thermal conductivity measurement while adopting the FDTR method as a validation. The measured thermal conductivity from the FDTR method matches trends and outcomes observed from the steady-state method.

Figure R9. Representative data and corresponding fitting curve in the plot of phase angle and frequency.

REVIEWER COMMENTS

Reviewer #1 (Remarks to the Author):

The authors have made commendable efforts to revise their manuscript in response to the initial round of reviews. However, there are still several critical points that need to be addressed:

1) Regarding the calculation of the time constant for material deformation due to mechanical stretching, the authors have employed the formula $\tau_{\text{strain}} = (\lambda - 1)L/V$, which is applicable to materials with linear stress-strain relationships. However, the polymer materials used in the study typically exhibit significant viscoelastic characteristics and nonlinear stress-strain behavior. The authors are requested to confirm the applicability of the employed formula and its specific impact on the time constant calculation.

2) The applicability of the photoelastic scattering experiment is called into question. The description of the experimental method in the manuscript may lack sufficient detail, particularly in explaining how the response time information is extracted from the photoelastic scattering experiments. Moreover, the use of this method to indirectly assess the structural response time may not fully simulate the actual thermal conductivity changes. The introduction of cracks during the experiment could significantly alter the stress state of the material, thereby affecting the accuracy of the experimental results. The authors are advised to discuss the limitations of this experimental approach in the revised manuscript and consider whether additional experiments are necessary to validate the response time. Additionally, the authors should clarify whether the response time of the thermal switch should be defined as the on-time or the on-off cyclical time.

3) In the performance comparison of thermal switches (Fig. 3), the authors have positioned the response time of ferroelectric material thermal switches at tens of seconds (Refs. 14 and 52), while the reviewer has noted that the cited literature reports subsecond response times, with one explicitly stating that "Although the response is clearly subsecond, in actuality, the switching most likely happens over much shorter time scales, possibly as short as 40 ns, as the absolute speed of detection is limited by the 300 ms time constant of the lock-in amplifier used in the TDTR measurement" (ACS Appl. Mater. Interfaces 2018, 10, 30, 25493). In fact, the latest report on antiferroelectric PbZrO₃ thin film thermal switches has documented a response time of ~150 ns (Science 2023, 382, 1265–1269). The inaccuracy or inconsistency in the authors' citation of data has led the reviewer to reserve judgment on the credibility of the study.

4) The reviewer also expresses reservations regarding the complexity and practical feasibility of the dual-field synergistic control of the thermal switch.

Reviewer #2 (Remarks to the Author):

I have gone through the comments raised from the other reviewers and mine, and the corresponding responses as well. The authors appear to have in general satisfactorily responded to the points raised in the previous reviews, and improved the manuscript. There does not appear to be a problem with publishing it.

Reviewer #3 (Remarks to the Author):

we recommend accepting this revised manuscript, as our concerns have been thoroughly addressed in a point-by-point manner. Prior to publication, we advise the authors to meticulously review the manuscript as there is room for further enhancement in its presentation. For instance, in the revised Figure 2d, the data bars representing the "Tuning ratio" are disproportionately positioned at the bottom of the figure, resulting in an awkward visual representation unless there is a specific intended effect.

Response to Review Comments (Manuscript ID: NCOMMS-23-42501B)
**“Reversible Two-way Tuning of Thermal Conductivity
in an End-linked Star-shaped Thermoset”**

Response to Reviewer 1

General comment. The authors have made commendable efforts to revise their manuscript in response to the initial round of reviews. However, there are still several critical points that need to be addressed.

Response to general comment. Thank you for acknowledging our efforts in response to the initial round of reviews. We also appreciate your new comments and suggestions, which further strengthen the paper. In the following sections, we address each comment point by point. Newly inserted text in the manuscript and supplementary information are marked in **red**.

Comment 1. Regarding the calculation of the time constant for material deformation due to mechanical stretching, the authors have employed the formula $\tau_{\text{strain}} = (\lambda - 1)L/V$, which is applicable to materials with linear stress-strain relationships. However, the polymer materials used in the study typically exhibit significant viscoelastic characteristics and nonlinear stress-strain behavior. The authors are requested to confirm the applicability of the employed formula and its specific impact on the time constant calculation.

Response to comment 1. Thank you for your comments. We would like to clarify that the formula $\tau_{\text{strain}} = (\lambda - 1)L/V$ is not exclusive to materials with linear stress-strain relationships. Instead, this formula applies to any material, regardless of stress-strain nonlinearity. The foundation of the formula is based on the definition of strain,

$$\varepsilon = \lambda - 1 = \Delta l/L \quad (1)$$

where λ is stretch, Δl is the length change of the specimen (or equivalently the loading displacement applied on the specimen), and L is the initial length of the specimen. The time for stretching is related to velocity V by

$$\tau_{\text{strain}} = \Delta l/V \quad (2)$$

Combining Eqs. (1-2), we can obtain $\tau_{\text{strain}} = (\lambda - 1)L/V$. Notably, Eqs. (1-2) do not require the stress-strain relationship of the material to be linear.

In addition, we would like to clarify that our ELST exhibits negligible viscoelasticity in the frequency operating range for thermal conductivity tuning (i.e., 0.1 ~ 10 Hz), which is intrinsically different from conventional polymers made of randomly crosslinked polymer networks. The negligible viscoelasticity of our ELST is achieved because it contains few topological defects and nearly no trapped chain entanglements, which manifests via its performance in dynamic mechanical analysis and stress-stretch measurements. We fabricated a randomly crosslinked end-linked polymer as a control with the same concentration and same polymer chemistry as our ELST to represent conventional polymers. As shown by the dynamic mechanical analysis (DMA) in **Fig. R1a**, the storage modulus (E') of our ELST above T_m (60 °C) remained constant with frequency

variation from 0.01 to 10 Hz, validating the negligible viscoelastic characteristics of our ELST. In contrast, the randomly crosslinked end-linked polymer exhibits significant rate dependence. Furthermore, our ELST exhibits much less mechanical hysteresis (**Fig. R1b**) compared to the randomly crosslinked end-linked polymer (**Fig. R1c**), further indicating our ELST exhibits negligible viscoelasticity. More details are referred to our recently published paper (C. Hartquist, et al., *Sci. Adv.*, 9, 50, 2023).

Comment 2-1. The applicability of the photoelastic scattering experiment is called into question. The description of the experimental method in the manuscript may lack sufficient detail, particularly in explaining how the response time information is extracted from the photoelastic scattering experiments.

Response to comment 2-1. Many thanks for the reviewer's detailed comments associated with the photoelasticity experiment. Specifically, we performed the relaxation photoelasticity experiment on a material subjected to an applied step strain (**Fig. R2a**), measuring the nominal stress of the material s and the intensity at specific locations on the material I as a function of time t . As illustrated in **Fig. R2b, c**, by fitting the measured $s(t)$ and $I(t)$ into the Voigt model¹, we can extract the mean response time for the structural change of the entire material τ_m and the response time for the structural change at specific locations on the material τ via the following equations,

$$\frac{s(t) - s_{min}}{s_{max} - s_{min}} = 1 - e^{t/\tau_m} \quad (3)$$

$$\frac{I(t) - I_{min}}{I_{max} - I_{min}} = 1 - e^{t/\tau} \quad (4)$$

where s_{max} and s_{min} are the maximum and minimum nominal stress of the material, and I_{max} and I_{min} are the maximum and minimum intensity at specific locations on the material. **Fig. R2d**

shows a representative force as a function of time, measuring τ_m ; **Fig. R2e** presents a representative color map as a function of time, measuring τ at different locations.

Figure R2. Illustration of extracting response time in the photoelasticity experiment. **a**, The material is subjected to an applied step strain. **b**, The recorded normal stress of the entire material s as a function of time t . **c**, The measured intensity at specific locations on the material I as a function of time. **d**, Representative force versus time in a PAAm gel subjected to an instantaneous change of step strain. **e**, Representative color map versus time in a PAAm gel subjected to an instantaneous change of step strain.

Revision for comment 2-1. To provide more details on extracting the response time in the photoelasticity experiment, we added a new paragraph in the Supporting Information, “Specifically, we performed relaxation photoelasticity experiments on our ELST (**Fig. S10a, b**), measuring the nominal stress of the material s and the intensity at specific locations on the material I as a function of time t . As illustrated in **Fig. S9b-d**, by fitting the measured $s(t)$ and $I(t)$ into the Voigt model¹, we can extract the mean response time for the structural change of the entire material τ_m and the response time for the structural change at specific locations on the material τ via the following equations, $\frac{s(t)-s_{min}}{s_{max}-s_{min}} = 1 - e^{-t/\tau_m}$, $\frac{I(t)-I_{min}}{I_{max}-I_{min}} = 1 - e^{-t/\tau}$, where s_{max} and s_{min} are the maximum and minimum nominal stress of the material, I_{max} and I_{min} are the maximum and minimum intensity at specific locations on the material. **Figure S10c** shows a representative force of our ELST as a function of time, which measures τ_m ; **Figure S10d** presents representative intensities at specific locations of our ELST as a function of time, which measures the corresponding τ at these locations. The value of τ_m and the mean value of spatially distributed τ are consistently on the order of 1 second, which indicates the response time for structure change in the sample is almost the same as the response time for stress relaxation of the entire sample. Physically, these two response times align with the response time for the thermal conductivity tuning since the polymer’s thermal conductivity is inherently linked to its molecular structure.”

Figure S10. Relaxation photoelasticimetry experiments. **a)** Schematic illustration of an instantaneous stretch applied on the sample followed by stress relaxation. **b)** Image sequences of fringe patterns in our ELST subjected to an instantaneous stretch λ of 4.4 along the horizontal direction followed by stress relaxation for up to 200s. **c)** Measured stress as a function of time fitted into the Voigt model to extract the mean response time for the structural change of the entire material, $\tau_m = 2.58$ s. **d)** Measured intensities at specific locations P₁, P₂ and P₃ as a function of time fitted into the Voigt model to extract the corresponding response time for the structural change at these locations $\tau_1 = 0.29$ s, $\tau_2 = 0.88$ s, and $\tau_3 = 0.63$ s. **e)** Comparison of the mean response time τ_m between a PAAm gel and our ELST.

In the main text line 239-243, we add “Specifically, from the measured nominal stress as a function of time, we extract the mean response time for the structural change of the entire sample as 2.58s (**Supplementary Fig. S10c**); from the measured intensities at specific locations as a function of time, we extract the response time for the structural change at these locations are consistently on the order of 1s (**Supplementary Fig. S10b, d**)”.

Comment 2-2. Moreover, the use of this method to indirectly assess the structural response time may not fully simulate the actual thermal conductivity changes. The introduction of cracks during the experiment could significantly alter the stress state of the material, thereby affecting the accuracy of the experimental results.

Response to comment 2-2. Many thanks for the reviewer’s comment. To increase light intensity for enhanced resolution through amplified stress levels, we introduced a crack at the center of the sample for inducing stress concentration. While we agree that the introduction of a crack might

potentially induce experimental error due to the altered stress state in the material, our experiments indicate the measured response time for structure changes in a material with a crack is almost the same as that in the same material with no crack, as manifested by **Fig. R3**. To further examine the accuracy of the experimental results due to the different stress levels applied on the material, we plot the intensity versus time at different locations away from crack tip. As shown in **Fig. R3b**, the extracted relaxation time measured on the ELST with a crack at different locations is consistently on the order of 1s, almost the same as the ELST without a crack (**Fig. R3a**).

In addition, we ascertain that the response time for structural change in the photoelasticimetry experiment should dictate the response time for thermal conductivity change. The thermal conductivity of a polymeric material is inherently determined by the material's molecular architecture. Once the material's molecular architecture reaches its steady state after the material is subjected to a strain or temperature variation, the material's thermal conductivity should reach its steady state as well.

Revision for comment 2-2. To clarify the accuracy of the response time for structure changes and its indication of the response time for thermal conductivity change, we provided more discussion in the Supporting Information, “To increase light intensity for enhanced resolution – which improves measurement accuracy – through amplified stress levels, we introduced a crack at the center of the sample for inducing stress concentration. While the introduction of the crack can enhance resolution, the presence of a crack could induce experimental error due to the altered stress state in the material. To examine the accuracy of the experimental results due to the different stress levels applied on the material, we plot the intensity versus time at different locations away from crack tip. As shown in **Fig. S10d**, the extracted relaxation time slightly decreases with the distance away from the crack tip but is consistent on the order of 1 s”.

Comment 2-3. The authors are advised to discuss the limitations of this experimental approach in the revised manuscript and consider whether additional experiments are necessary to validate the response time.

Response to comment 2-3. Thanks for the reviewer’s comment. The relaxation photoelasticity experiment has two primary limitations. The first limitation lies in its indirect measurement nature, introducing potential sources of error. For example, the resolution of the measured response time is dependent on the optical system’s ability to detect the color change of our ELST. This detail could account for differences in relaxation time observed in the relaxation photoelasticity experiment (**Fig. S10d**) compared to the stress relaxation experiment (**Fig S10c**). The second limitation is that this technique only works for optically transparent or translucent materials. Fortunately, our ELST becomes optically transparent when heated above its melting temperature, enabling effective use of the relaxation photoelasticity experiment.

We also explored an alternative experiment aimed at directly measuring the response time, using a thermal camera. The working principle is to detect temperature changes resulting from variations in thermal conductivity within the material under strain or subjected to temperature variations. However, technical challenges arise as our ELST needs to be elastically stretched above its melting temperature (60 °C), which requires either an environmental chamber or an oil bath for temperature control. Using an oil bath is impractical because the thermal camera would measure the oil surrounding the sample instead of the sample itself. Employing an environmental chamber presents technical difficulties, because instantaneous changes in strain or temperature applied to the material significantly disturbs the temperature in the chamber, therefore the measured response time is significantly impacted by the time required for the environment chamber to reach its temperature equilibrium. Consequently, we have determined that the relaxation photoelasticity experiment complemented by the stress relaxation experiments remains the most suitable method to measure the response time for structure change, indirectly probing the response time for thermal conductivity change.

Revision for comment 2-3. In the revised Supplementary Information, we provide discussion on the limitations of the relaxation photoelasticity experiment, adding that “**It should be noted that the relaxation photoelasticity experiment has two primary limitations. The first limitation lies in its indirect measurement nature, introducing potential sources of error. For example, the resolution of the measured response time is dependent on the optical system’s ability to detect the color change of our ELST. The second limitation is that this technique only works for optically transparent or translucent materials. Fortunately, our ELST becomes optically transparent when heated above its melting temperature, enabling effective use of the relaxation photoelasticity experiment.**”

Comment 2-4. Additionally, the authors should clarify whether the response time of the thermal switch should be defined as the on-time or the on-off cyclical time.

Response to comment 2-4. Many thanks for the reviewer’s great comment. The response time of the thermal switch is the on-time. As shown in **Fig. R4c**, we performed new cyclic tests to measure

both the on and off tuning time as 2.58 s and 0.08 s, respectively. Notably, the time for off tuning is shorter compared to on tuning, which is possibly due to minimal alterations in the polymer-network architecture when our ELST is unloaded. (It should be noted that the off tuning time is measured using the stress relaxation test instead of the photoelasticimetry relaxation test, which is technically challenging primarily due to the negligible photoelasticity of our ELST at its undeformed state.) Additionally, our data suggest that the one-cycle on and off tuning time remain nearly identical across various cycles (Fig. R4b).

Revision for comment 2-4. Since most references from the literature report the on-time as thermal conductivity tuning time, we similarly take the on-time as thermal conductivity tuning time throughout the manuscript. In addition, we perform cyclic relaxation experiments to measure the on-time and off-time over 10 cycles, as shown in Fig. R4.

In addition, in the main text line 249-253, we add “We also performed the cyclic relaxation experiments (Supplementary Fig. S11) to measure both the on and off tuning time as 2.58 s and 0.08 s, respectively. Notably, the time for off tuning is shorter compared to on tuning, which is possibly due to minimal alterations in the polymer-network architecture when our ELST is unloaded. Additionally, our data suggest that the on and off tuning time remain nearly identical across various cycles”.

Comment 3. In the performance comparison of thermal switches (Fig. 3), the authors have positioned the response time of ferroelectric material thermal switches at tens of seconds (Refs. 14 and 52), while the reviewer has noted that the cited literature reports subsecond response times, with one explicitly stating that "Although the response is clearly subsecond, in actuality, the switching most likely happens over much shorter time scales, possibly as short as 40 ns, as the absolute speed of detection is limited by the 300 ms time constant of the lock-in amplifier used in the TDTR measurement" (ACS Appl. Mater. Interfaces 2018, 10, 30, 25493). In fact, the latest report on antiferroelectric PbZrO₃ thin film thermal switches has documented a response time of

~150 ns (Science 2023, 382, 1265–1269). The inaccuracy or inconsistency in the authors' citation of data has led the reviewer to reserve judgment on the credibility of the study.

Response to comment 3. We very appreciate the review's careful checking. In our response to the initial round of revision, we collected the data from the figures in the papers, which lead to inaccuracy. We have carefully gone through all cited literature in **Fig. 3** and **Table S1** and corrected accordingly. It should be noted that, in addition to the exact value that is explicitly reported in the literature, we use a shaded region to indicate the range of the response time. For example, for the latest report on antiferroelectric PbZrO₃ thin film thermal switches (Science 2023, 382, 1265–1269), we did include its response time on the order of 100 ns. Furthermore, we need to highlight that the correction does not diminish the novelty of our ELST in outperforming existing polymeric thermal switches due to its unique reversible and significant thermal conductivity tuning capabilities.

Revision for comment 3. We report **Fig. 3** in the Main Text as shown below and update the form in the **Supplementary Table S1**.

Comment 4. The reviewer also expresses reservations regarding the complexity and practical feasibility of the dual-field synergistic control of the thermal switch.

Response to comment 4. Thank you for the reviewer's comment. As shown in **Fig. R6**, our ELST's two-way thermal conductivity tuning primarily relies on strain modulation, with thermal modulation acting as an additional tuning capacity. In particular, we would like to highlight that single-field control (i.e., strain control) of our ELST can already achieve approximately 80% of the maximum tuning capacity with $\sim 11.5 \times$ thermal conductivity tuning ratio. In practical

applications, utilizing strain modulation as the primary control while employing thermal modulation as supplementary tuning is both reasonable and feasible.

Moreover, our ELST demonstrates superior elastic stretchability and ultra-high strain-induced crystallization, showcasing its potential for solid-state elastocaloric cooling as demonstrated in our paper (C. Hartquist, et al., *Sci. Adv.*, 9, 50, 2023). The capability to adjust the ELST's thermal conductivity under controlled strain and temperature conditions holds promise for maximizing adiabatic temperature changes by minimizing thermal losses to the environment.

While most thermal switches respond to a single stimulus, our ELST unveils avenues to explore how responses to multiple stimuli could introduce a new paradigm. The synergy between these stimuli could potentially yield unprecedented results.

Looking ahead, future efforts to enhance the feasibility of the ELST could focus on molecular design strategies aimed at manipulating the material's melting temperature to align with the working temperature for optimal performance.

Revision for comment 4. To clarify the practical feasibility of our ELST, in line 181 and 189, we added “It is important to highlight that the two-way thermal conductivity tuning of our ELST is primarily driven by strain modulation, with thermal modulation serving as an additional tuning capacity. Notably, employing single-field control (i.e., strain control) of our ELST can already achieve approximately 80% of the maximum tuning capacity, resulting in an $11.5\times$ thermal conductivity tuning ratio. This underscores the feasibility of utilizing strain modulation as the primary control, complemented by thermal modulation for supplementary tuning, making the ELST well-suited for practical applications. While most thermal switches respond to a single stimulus, our ELST unveils avenues to explore how responses to multiple stimuli could introduce a new paradigm, which could potentially yield unprecedented results.” in the main text.

Figure R6. Effects of strain and temperature on thermal conductivity of ELST. a, Thermal conductivity of the ELST as a function of stretch ratio at $T = 30$ and $60\text{ }^\circ\text{C}$. b, Thermal conductivity of the ELST as a function of temperature at $\lambda = 1$ and 20 .

Response to Reviewer 2

General comment. I have gone through the comments raised from the other reviewers and mine, and the corresponding responses as well. The authors appear to have in general satisfactorily responded to the points raised in the previous reviews and improved the manuscript. There does not appear to be a problem with publishing it.

Response to general comment. Thank you for acknowledging our efforts in revision and recommending its publication.

Response to Reviewer 3

General comment. We recommend accepting this revised manuscript, as our concerns have been thoroughly addressed in a point-by-point manner.

Response to general comment. Thank you for acknowledging our efforts in revision and recommending its publication.

Comment 1. Prior to publication, we advise the authors to meticulously review the manuscript as there is room for further enhancement in its presentation. For instance, in the revised Figure 2d, the data bars representing the "Tuning ratio" are disproportionately positioned at the bottom of the figure, resulting in an awkward visual representation unless there is a specific intended effect.

Response to comment 1. Many thanks for the comments. We have gone through all the key figures to enhance their presentations. However, we would like to clarify that we intentionally scaled the tuning ratio in **Fig. 2d** matching that in **Fig. 2c** to highlight the two-way thermal conductivity tuning of our ELST is dominated by strain modulation while thermal modulation serves as an additional tuning capacity. Notably, we place the two panels measuring the tuning ratios side-by-side to mark the intended effect of their scaling.

Revision for comment 1. We reorganize **Fig. 2** as shown below to further enhance its presentation and avoid potential confusion.

REVIEWERS' COMMENTS

Reviewer #1 (Remarks to the Author):

The authors have effectively addressed all my previous concerns, further strengthening the manuscript's contribution to the field. I recommend publication without further revisions.

Response to Review Comments (Manuscript ID: NCOMMS-23-42501B)
**“Reversible Two-way Tuning of Thermal Conductivity
in an End-linked Star-shaped Thermoset”**

Response to Reviewer 1

General comment. The authors have effectively addressed all my previous concerns, further strengthening the manuscript's contribution to the field. I recommend publication without further revisions.

Response to general comment. Thank you for acknowledging our efforts in revision and recommending its publication.